# Weakened APC/C activity at mitotic exit drives cancer vulnerability to KIF18A inhibition

Colin R Gliech [1], Zhong Y Yeow[1], Daniel Tapias-Gomez[1], Yuchen Yang [1], Zhaoyu Huang[1], Andréa E Tijhuis [2], Diana CJ Spierings [2], Floris Foijer [2], Grace Chung[3], Nuria Tamayo [4], Zahra Bahrami-Nejad[5], Patrick Collins[5], Thong T Nguyen[5], Andres Plata Stapper [6], Paul E Hughes [3], Marc Payton[3] & Andrew J Holland [1][✉]

## Abstract

**The efficacy of current antimitotic cancer drugs is limited by toxicity in highly proliferative healthy tissues. A cancer-specific dependency on the microtubule motor protein KIF18A therefore makes it an attractive therapeutic target. Not all cancers require KIF18A, however, and the determinants underlying this distinction remain unclear. Here, we show that KIF18A inhibition drives a modest and widespread increase in spindle assembly checkpoint (SAC) signaling from kinetochores which can result in lethal mitotic delays. Whether cells arrest in mitosis depends on the robustness of the metaphase-to-anaphase transition, and cells predisposed with weak basal anaphase-promoting complex/cyclosome (APC/C) activity and/or persistent SAC signaling through metaphase are uniquely sensitive to KIF18A inhibition. KIF18A-dependent cancer cells exhibit hallmarks of this SAC:APC/C imbalance, including a long metaphase-to-anaphase transition, and slow mitosis overall. Together, our data reveal vulnerabilities in the cell division apparatus of cancer cells that can be exploited for therapeutic benefit.**

**Keywords** Anaphase Promoting Complex (APC/C); Cancer; KIF18A; Mitosis; Spindle Assembly Checkpoint (SAC)
**Subject Categories** Cancer; Cell Cycle

## Introduction

Cancer is characterized by uncontrolled cell growth. Consequently, the cell cycle and mitosis have long been appealing targets for chemotherapy. Early successes with "antimitotic" drugs emerged from using microtubule targeting agents such as taxanes and vinca alkaloids to disrupt the assembly of the mitotic spindle apparatus and kill dividing cells (Checchi et al, 2003; Jordan and Wilson, 2004). Despite decades of widespread usage and efficacy across several tumor types, these drugs suffer from significant limitations: antimitotic drugs also kill healthy, highly proliferative cells of the bone marrow and gut, and significant neurotoxicity is observed due to the disruption of microtubule dynamics in nondividing neurons (Čermák et al, 2020; Gornstein and Schwarz, 2014; Klein and Lehmann, 2021; Marupudi et al, 2007; Shemesh and Spira, 2009). These drawbacks fueled the development of a new class of drugs that target essential mitotic proteins such as kinases and microtubule motor proteins. This second generation of antimitotic agents mitigates the neurotoxicity observed with microtubule targeting agents. However, myelosuppression remains a dose-limiting toxicity in patients (Chan et al, 2012; Serrano-del Valle et al, 2021), and as a result, these drugs have so far failed to progress in the clinic.

Because antimitotic agents agnostically kill all dividing cells, identifying and exploiting tumor-specific vulnerabilities in the cell division apparatus remains a major opportunity for drug development. One promising target is the plus-end-directed motor protein KIF18A, which is uniquely essential for the division of certain cancers (Cohen-Sharir et al, 2021; Marquis et al, 2021; Quinton et al, 2021). KIF18A accumulates at the plus ends of kinetochore microtubules and suppresses chromosome oscillations within the metaphase plate (Du et al, 2010; Mayr et al, 2007; Stumpff et al, 2008; Zhu et al, 2005). Loss of KIF18A disrupts chromosome alignment, leading to an increased instance of lagging anaphase chromosomes and micronucleation (Fonseca et al, 2019; Lin et al, 2020; Quinton et al, 2021). KIF18A loss has been shown to activate the spindle assembly checkpoint (SAC) at microtubule-attached kinetochores (Janssen et al, 2018), suggesting incomplete microtubule occupancy or tension defects. Despite roles in chromosome alignment and SAC silencing, KIF18A is largely dispensable for accurate chromosome segregation in non-transformed cells (Fonseca et al, 2019), and mice with knockout or dysfunctional KIF18A survive through adulthood (Czechanski et al, 2015; Liu et al, 2010; Sepaniac et al, 2021).

[1]Department of Molecular Biology and Genetics, Johns Hopkins University School of Medicine, Baltimore, MD 21205, USA. [2]European Research Institute for the Biology of Ageing, University of Groningen, University Medical Center Groningen, Groningen, AV 9713, The Netherlands. [3]Oncology Research, Amgen Research, Thousand Oaks, CA 91320, USA. [4]Medicinal Chemistry, Amgen Research, Thousand Oaks, CA 91320, USA. [5]Genome Analysis Unit, Amgen Research, South San Francisco, CA 94084, USA. [6]Center for Research Acceleration by Digital Innovation, Amgen Research, South San Francisco, CA 94084, USA. [✉]E-mail: aholland@jhmi.edu

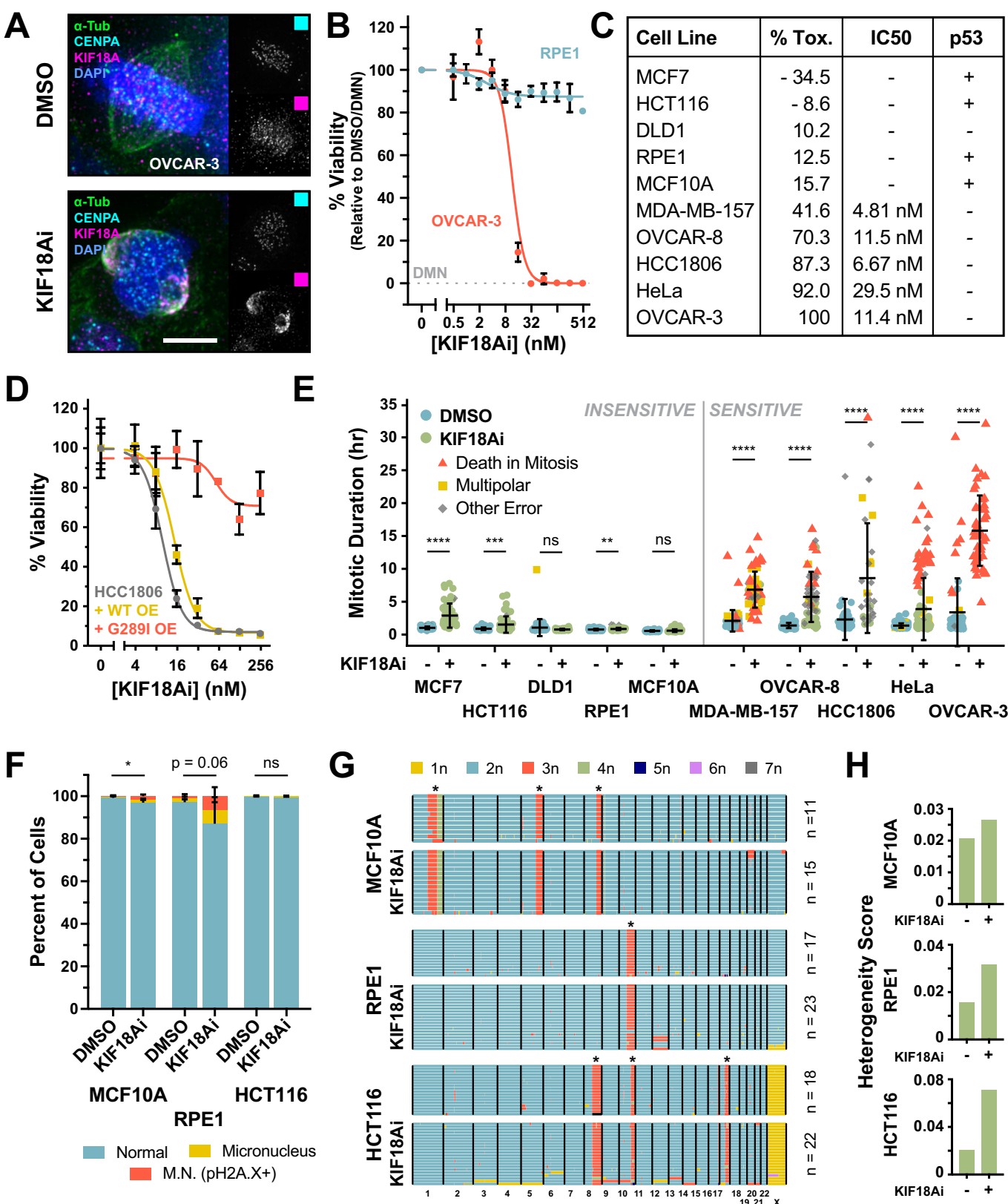

**Figure 1. Sensitivity to KIF18A inhibition is defined by long mitotic delays that drive catastrophic errors.**

(A) Wide-field immunofluorescence of KIF18A localization relative to kinetochores (CENP-A, cyan) and the mitotic spindle (α-Tubulin, green) in DMSO or KIF18Ai-treated OVCAR-3 cells. Scale bar = 5 μm. (B) Titration of KIF18Ai in 5-day MTT endpoint viability assay for RPE1 and OVCAR-3 cells. N = 4 independent experiments, n ≥ 2 technical replicates per experiment. Data are represented as mean ± SEM. (C) Summary table of KIF18Ai toxicity, $IC_{50}$ values and p53 pathway functionality from MTT assays in Fig. EV1B,E. Toxicity values are derived from plateau measurements of the $IC_{50}$ dose–response curves. (D) Titration of KIF18Ai in 5-day MTT endpoint viability assay for HCC1806 cells constitutively expressing a WT or drug-resistant (G289I) KIF18A-(3x)HA transgene. Data are represented as mean ± SD. N = 3 technical replicates from a single experiment. (E) Quantification of live-cell wide-field time-lapse microscopy of H2B/α-Tubulin fluorescently tagged cell lines colored by mitotic outcome. Data are represented as mean ± SD. n > 35 cells per condition. Statistical significance was determined using an unpaired two-tailed Student's *t* test. Sample size and full statistical results are listed in Dataset EV2. (F) Quantification of micronucleus formation after 7 days of DMSO or KIF18Ai treatment in indicated cell lines. Phospho-histone H2A.X+ micronuclei denote DNA damage and likely micronucleus rupture. Data are represented as mean ± SD. N = 5 independent experiments, n ≥ 100 cells per condition per experiment. Statistical significance was determined using an unpaired two-tailed Student's *t* test. Sample size and full statistical results are listed in Dataset EV2. (G) scDNAseq of indicated cell lines following 7 days of DMSO or KIF18Ai treatment. The asterisk indicates common karyotypic alterations in each cell line. (H) Heterogeneity score calculation for each sample in (F). Data Information: *P < 0.05, **P < 0.01, ***P < 0.001, and ****P < 0.0001 (E, F). Source data are available online for this figure.

KIF18A inhibitor treatment across the large-scale pan-cancer PRISM panel (629 cell lines) revealed that 25% of cancers overall and 32% of ovarian cancers exhibited strong KIF18A dependency (Payton et al, 2023). It remains unclear, however, what features distinguish KIF18A-dependent and agnostic cancer cells. Chromosomal instability (CIN) is a common feature of human tumors that results in high rates of chromosome mis-segregation during cell division. It has been proposed that the altered microtubule dynamics in cancers with CIN render them more reliant on KIF18A function (Marquis et al, 2021). Separately, whole-genome doubling (WGD) and aneuploidy have been shown to increase KIF18A dependency (Cohen-Sharir et al, 2021; Quinton et al, 2021). Since WGD promotes CIN and aneuploidy (Prasad et al, 2022), the sensitization mechanisms reported in these studies may be interrelated. Nevertheless, neither WGD nor CIN fully explains sensitivity to KIF18A loss, and we lack a unified mechanistic model for KIF18A dependency.

Here, using a small molecule inhibitor of KIF18A, we demonstrate that long mitotic delays and lethal errors in cell division drive cell death in KIF18A-dependent cancers. KIF18A inhibition halts cells in mitosis by destabilizing kinetochore–microtubule interactions leading to modest SAC activation across most kinetochores. This increase in SAC signal is common among KIF18A-dependent and agnostic cancers but is exacerbated by increased ploidy. The anaphase-promoting complex (APC/C) is inhibited by the SAC to prevent mitotic exit, and we show cells with low basal APC/C activity or those that do not fully silence SAC signaling at metaphase are particularly sensitive to further SAC activation with KIF18A inhibition. Collectively, our results reveal multiple features contributing to KIF18A dependency that can be used to guide the development of clinical indicators for the effective use of KIF18A inhibitors.

## Results

### Sensitivity to KIF18A inhibition is defined by long mitotic delays that drive catastrophic mitotic errors

To investigate the determinants of KIF18A dependency in cancer, we used AM-1882, a highly specific small molecule KIF18A inhibitor (KIF18Ai) developed by Amgen Inc (Payton et al, 2023; Tamayo et al, 2022). Treatment with KIF18Ai causes stable binding of the KIF18A motor domain to microtubules (Tamayo et al, 2022), leading to the relocation of KIF18A in mitosis from kinetochores to

spindle poles in all the cell lines tested (Figs. 1A and EV1A). This mis-localization phenocopies the mitotic effects of KIF18A depletion (Payton et al, 2023), likely by preventing KIF18A-induced stabilization of microtubule plus ends at the metaphase plate (Du et al, 2010). We examined the effect of KIF18Ai on cell viability using a 5-day endpoint assay performed with non-transformed cells (RPE1 and MCF10A) and a panel of the colon (HCT116, DLD1), breast (MCF7, MDA-MB-157, HCC1806), ovarian (OVCAR-3, OVCAR-8), and cervical (HeLa) cancer cell lines (Figs. 1B,C and EV1B), which were selected from a previously published large-scale drug sensitivity screen (Payton et al, 2023). Sensitivity to KIF18Ai varied across the cell line panel and did not correlate with KIF18A expression levels (Fig. EV1C,D). In accordance with previous work, p53 loss of function was common in cell lines sensitive to KIF18A inhibition, though the functional loss of the p53 pathway could not explain high inhibitor toxicity (Figs. 1C and EV1E; Dataset EV1). For downstream analysis, we divided cell lines into KIF18Ai-sensitive or -insensitive groups: sensitive cell lines exhibited >40% toxicity to KIF18Ai, while insensitive cell lines showed <20% toxicity over a 5-day period.

To confirm that the toxicity from KIF18Ai treatment was due to impaired KIF18A function, we set out to generate a drug-resistant mutant of KIF18A. KIF18Ai has partial activity toward the related kinesin KIF19 but not KIF18B (Payton et al, 2023). Exploiting this distinction, we identified a drug-facing glycine present in the KIF18Ai binding pocket of KIF18A and KIF19 that is changed to isoleucine in KIF18B (Fig. EV2A) (Locke et al, 2017; Tamayo et al, 2022; Waterhouse et al, 2009). We hypothesized that this G:I substitution was the cause of the KIF18Ai resistance for KIF18B. Consistently, overexpression of a KIF18A G289I transgene, but not WT KIF18A, dramatically relieved KIF18Ai toxicity (Figs. 1D and EV2B,C). This demonstrates that KIF18Ai toxicity results from the loss of KIF18A function.

We next evaluated the immediate effects of KIF18A inhibition on mitosis using live-cell imaging of cells labeled with a fluorescent histone H2B and α-Tubulin. While insensitive cells divided mostly normally with KIF18Ai treatment, sensitive cell lines experienced varying degrees of lengthy mitotic delays that often resulted in lethal cell division errors (Fig. 1E; Movies EV1–EV5). Consistent with KIF18A depletion experiments (Stumpff et al, 2008), KIF18Ai treatment also led to the appearance of severely misaligned chromosomes. KIF18Ai-sensitive cell lines preferentially underwent either multipolar divisions (MDA-MB-157), chromosome segregation errors (HCC1806), or death in mitosis (MDA-MB-157, OVCAR-8, HeLa, OVCAR-3) (Fig. EV3A). Sensitive cell lines also

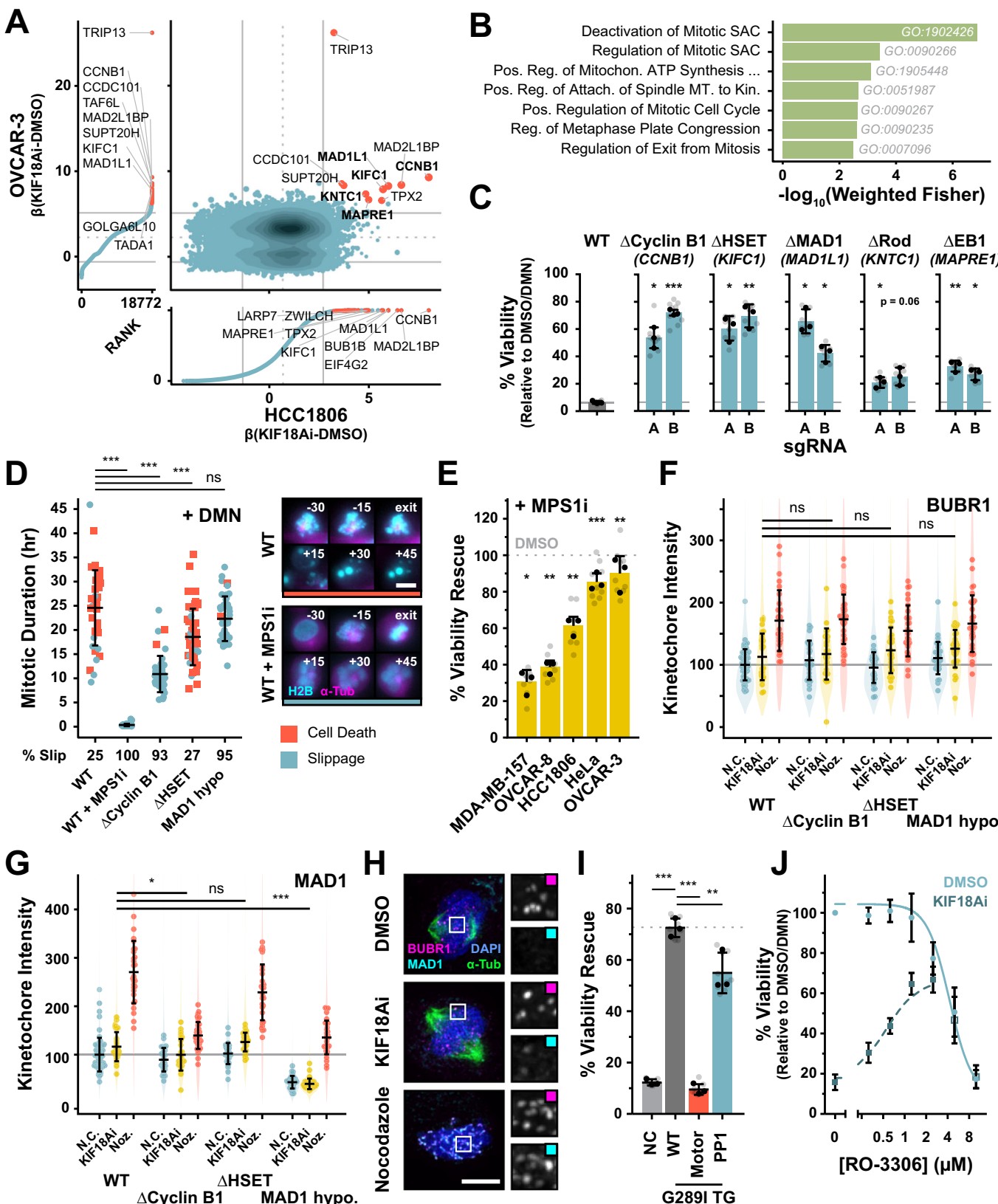

**Figure 2. SAC activation drives KIF18Ai toxicity.**

(A) Comparison of OVCAR-3 and HCC1806 whole-genome CRISPR–Cas9 screens. Data is annotated with mean ± 1.5 × SD. Hits with an average FDR < 0.3 are highlighted in red. (B) Gene set enrichment analysis of genes from (A) with a β(KIF18Ai-DMSO) > 1.5 SD and an average FDR < 0.3. The significance of each GO term was evaluated with a Fisher's exact test using the topGO R package. (C) 5-day MTT endpoint viability assay of HCC1806 polyclonal CRISPR–Cas9 KO cell lines in KIF18Ai. Data are represented as mean ± SD. $N = 3$ independent experiments, $n = 3$ technical replicates per experiment. Statistical significance was determined using a one-way ANOVA with post hoc Dunnett's multiple comparisons test between each gene knockout pair and WT. Full statistical results are listed in Dataset EV2. (D) Quantification of mitotic duration and fate of mitotic HCC1806 monoclonal cell lines arrested with DMN. Data are represented as mean ± SD. $n \geq 40$ cells per condition. Statistical significance was determined using a one-way-ANOVA with post hoc Dunnett's multiple comparisons test between each sample and WT. Full statistical results are listed in Dataset EV2. Scale bar = 10 µm. (E) KIF18Ai viability rescue across sensitive cell line panel with sub-saturating Reversine treatment (30 nM) in a 5-day MTT endpoint assay. Data are represented as mean ± SD. $N = 3$ independent experiments, $n = 3$ technical replicates per experiment. Statistical significance was determined using a one-sample Student's $t$ test, $H_0$: % rescue = 0. Full statistical results are listed in Dataset EV2. (F) Quantification of BUBR1 intensity at kinetochores across HCC1806 clonal rescue cell lines in DMSO, KIF18Ai, and Nocodazole. Violin plots summarize all individual kinetochores analyzed, points represent per-cell intensity averages, and error bars are mean ± SD of per-cell averages. $N > 25$ cells per condition, $n > 2200$ kinetochores per condition. Statistical significance was determined using a one-way-ANOVA with post hoc Dunnett's multiple comparisons test between KIF18Ai intensity in edited cell lines and KIF18Ai intensity in WT. The sample size and full statistical results are listed in Dataset EV2. (G) Quantification of MAD1 intensity at kinetochores as in (F). $N > 25$ cells per condition, $n > 2200$ kinetochores per condition. Statistical significance was determined using a one-way ANOVA with post hoc Dunnett's multiple comparisons test between KIF18Ai intensity in edited cell lines and KIF18Ai intensity in WT. The sample size and full statistical results are listed in Dataset EV2. (H) Wide-field immunofluorescence images of SAC proteins BUBR1 and MAD1 at kinetochores in DMSO, KIF18Ai, and Nocodazole treatments in mitotic HCC1806 cells. Scale bar = 5 µm. (I) 5-day MTT endpoint viability assay of HCC1806 G289I KIF18A mutant overexpression cell lines in KIF18Ai. Data are represented as mean ± SD. $N = 3$ independent experiments, $n = 3$ technical replicates per experiment. Statistical significance was determined using a one-way-ANOVA with post hoc Dunnett's multiple comparisons test between each sample and G289I KIF18A (WT) overexpression. Full statistical results are listed in Dataset EV2. (J) Titration of CDK1 inhibitor RO-3306 in a 5-day MTT endpoint viability assay against DMSO or KIF18Ai-treated HCC1806 cells. $N = 3$ independent experiments, $n = 3$ technical replicates per experiment. Error bars represent mean ± SD. Data information: *$P < 0.05$, **$P < 0.01$, and ***$P < 0.001$ (C–E, G, I). Source data are available online for this figure.

exhibited a narrow threshold of mitotic duration (1.75–4 h) past which degeneration of the mitotic spindle or a loss of chromosome cohesion occurred (Fig. EV3B). Treatment with KIF18Ai extended the duration of mitosis past this threshold leading to spindle degeneration and/or cohesion loss and chronic SAC activation. By contrast, insensitive cell lines treated with KIF18Ai typically passed through mitosis with normal timing or, in the case of MCF7 cells, maintained spindle integrity and chromosome cohesion throughout a modestly prolonged division (Fig. 1E). This suggests that both an increased incidence of, and sensitivity to, mitotic delays are responsible for KIF18Ai toxicity.

Loss of KIF18A has previously been shown to cause low rates of chromosome mis-segregation and micronucleus formation (Lin et al, 2020; Quinton et al, 2021). We therefore investigated whether KIF18Ai-insensitive cell lines also suffered from these defects. A 7-day treatment with KIF18Ai modestly increased the proportion of micronuclei in insensitive RPE1 and MCF10A, but not HCT116 cells (Fig. 1F). We then used whole-cell, single-cell DNA sequencing to monitor changes in karyotype, where a moderate increase in aneuploidy was observed after KIF18Ai treatment in all three cell lines (Fig. 1G,H). We conclude that KIF18Ai treatment leads to mitotic delay and catastrophic mitotic errors in sensitive cell lines. By contrast, KIF18A inhibition in insensitive cell lines results in low rates of chromosome mis-segregation that have a minimal impact on short-term proliferation.

## CRISPR–Cas9 screens identify multiple pathways for KIF18Ai resistance

To identify regulators of KIF18A dependency, we performed paired genome-wide CRISPR–Cas9 knockout screens in two highly KIF18Ai-sensitive cell lines: HCC1806 and OVCAR-3. Cas9-expressing HCC1806 and OVCAR-3 cells were transduced with the Brunello sgRNA knockout library and selected for 7 days. Knockout HCC1806 and OVCAR-3 cells were grown in DMSO or KIF18Ai for 21 or 40 days, respectively, and guides that conferred resistance to KIF18Ai were identified by deep sequencing

(Appendix Fig. S1A; Dataset EV4). We predicted that two classes of genes would drive resistance to KIF18Ai: genes whose loss reduced the barrier to mitotic exit, and genes whose loss directly corrected the mitotic defects caused by KIF18A inhibition. In accordance with this expectation, SAC components and microtubule regulators were highly enriched in the screen hits (Fig. 2A,B). We focused on five hits with roles in the SAC (MAD1, Rod), mitotic spindle dynamics (HSET, EB1), or both (Cyclin B1). Independent polyclonal knockout HCC1806 cells expressing the top-performing sgRNAs for these genes demonstrated strong KIF18Ai resistance (>60%) for MAD1, HSET, and Cyclin B1 and modest resistance (<30%) for Rod and EB1 (Fig. 2C). We, therefore, focused on defining the role of MAD1, HSET, and Cyclin B1 loss in conferring KIF18Ai resistance.

Since residual protein was detectable in the polyclonal sgRNA-expressing populations (Appendix Fig. S1B), we derived clonal cell lines edited for Cyclin B1, MAD1, and HSET. Though we isolated multiple clones without detectable Cyclin B1 or HSET protein, all MAD1 clones retained low levels of MAD1 protein expression, suggesting that a complete loss of MAD1 is lethal in HCC1806 cells (Appendix Fig. S1C). In all cases, the viability rescue in KIF18Ai correlated well with the degree of Cyclin B1, MAD1, or HSET protein loss detected by western blot (Appendix Fig. S1D). In addition, live-cell time-lapse imaging demonstrated a reduced frequency of severe mitotic errors in response to KIF18Ai treatment in the clonal MAD1, HSET, and Cyclin B1-edited cell lines (Appendix Fig. S1E,F).

## SAC activation drives KIF18Ai toxicity

We predicted that depletion of MAD1 or Cyclin B1 might lower the barrier to mitotic exit and relieving KIF18Ai toxicity either by interfering with SAC signaling or by reducing the activity of the master mitotic kinase CDK1 (Brito and Rieder, 2006; Murray, 2004). To define the capacity of KIF18Ai-resistant HCC1806 clonal lines to maintain the mitotic state, we arrested cells in mitosis with the Eg5 inhibitor dimethylenastron (DMN) which causes spindle

collapse and induces chronic SAC activation (Collin et al, 2013; Müller et al, 2006). We then monitored mitotic slippage by live-cell time-lapse imaging. DMN treated WT HCC1806 cells stayed in mitosis for an average of 25 h, resulting in 75% cell death and 25% mitotic slippage (Fig. 2D). By contrast, abolishing SAC function with the MPS1 inhibitor Reversine led to all cells exiting mitosis in less than an hour. Hypomorphic MAD1 cells spent slightly less time in mitosis (23 h) and had a dramatically increased rate of mitotic slippage (95%). Knockout of Cyclin B1 reduced the time spent in mitosis to 11 h with similarly high slippage rates (93%), while HSET knockout cell lines spent slightly less time in mitosis (19 h) but had similar rates of mitotic slippage (27%) compared to control HCC1806 cells. Taken together, these data show that reductions in the level of Cyclin B1 or MAD1 suppress the ability of cells to delay mitosis and induce cell death.

We next sought to directly interrogate the role of the SAC on KIF18Ai toxicity. To test whether SAC suppression was sufficient to relieve KIF18A dependency, we co-treated the panel of sensitive cell lines with both KIF18Ai and low doses of Reversine. Partial SAC silencing rescued KIF18Ai toxicity across the panel of sensitive cell lines (Fig. 2E), with the extent of growth rescue likely limited by the intrinsic sensitivity to SAC inhibition following Reversine treatment (Appendix Fig. S2A). We conclude that KIF18Ai toxicity depends on SAC activation to induce mitotic delays.

Since cells with depleted Cyclin B1 or MAD1 were more prone to mitotic slippage, we speculated that they may have a reduced capacity to signal through the SAC. Several SAC proteins including BUBR1 and MAD1 are recruited to unattached kinetochores to execute SAC signaling and are displaced during SAC silencing (Chen et al, 1998; Taylor et al, 2001; Vigneron et al, 2004). We therefore employed fixed immunofluorescence to interrogate the degree of SAC signaling with KIF18Ai treatment relative to SAC off (metaphase, vehicle-treated) and SAC on (Nocodazole-treated) conditions. Unexpectedly, rather than resulting in strong SAC signaling at a few problematic kinetochores, KIF18Ai caused a minor increase in BUBR1 and MAD1 intensity across nearly all kinetochores (Fig. 2F–H). Clonal Cyclin B1, MAD1 and HSET edited lines all exhibited a similar capacity to recruit BUBR1 to kinetochores in response to KIF18Ai, suggesting that these cells remained proficient in sensing the KIF18Ai-driven defect (Fig. 2F). As expected, MAD1 signal was greatly diminished in MAD1 hypomorphic cells across all treatment conditions. However, we also observed slightly diminished MAD1 signal in Cyclin B1, but not HSET knockout cells (Fig. 2G). We conclude that depletion of Cyclin B1 or MAD1 alleviates KIF18Ai toxicity by reducing the cells' ability to signal through the SAC.

Displacement of SAC proteins at the kinetochore is driven by phosphatases PP2A-B56 and PP1 (Moura et al, 2017; Nijenhuis et al, 2014), and KIF18A harbors a regulatory PP1-binding motif that has been proposed to serve as a kinetochore recruitment platform for the phosphatase (De Wever et al, 2014; Häfner et al, 2014). We therefore speculated that elevated SAC signal with KIF18Ai may be a result of preventing KIF18A-mediated delivery of PP1 to direct SAC silencing. Using the drug-resistant KIF18A transgene system in HCC1806 cells, we generated a KIF18A variant lacking the PP1-binding motif (De Wever et al, 2014). As anticipated, WT G289I KIF18A, but not a similar transgene lacking motor activity (Stumpff et al, 2008), relieved drug toxicity. However, a G289I KIF18A mutant unable to bind PP1 largely

rescued growth (Figs. 2I and EV2B,C). We conclude that direct PP1 recruitment to the kinetochore by KIF18A does not play a major role in SAC silencing or KIF18Ai toxicity.

Since Cyclin B1 knockout only yielded a minor SAC signaling defect but provided a potent rescue of KIF18A toxicity, we wondered whether Cyclin B1 loss could be rescuing viability through other means. Notably, Cyclin B1 is critical for the activation of the master mitotic kinase CDK1 whose activity defines the mitotic state (Hayward et al, 2019; Yu and Yao, 2008). Accordingly, sub-saturating concentrations of the CDK1 inhibitor RO-3306 provided a robust rescue of viability in KIF18Ai-treated HCC1806 cells, suggesting high Cyclin B1/CDK1 activity is needed to maintain a mitotic arrest and promote KIF18Ai toxicity (Fig. 2J). Furthermore, Cyclin B1 serves a CDK1-independent scaffolding role for MAD1 at the kinetochore corona (Allan et al, 2020), a proteinaceous matrix that expands off of chronically unattached kinetochores to bolster microtubule capture and SAC signaling (Kops and Gassmann, 2020). However, ablation of the corona pool of MAD1 with targeted mutations in the protein had little impact on KIF18Ai resistance in HeLa cells (Appendix Fig. S2B) (Allan et al, 2020). These findings agree with the comparatively weak rescue of KIF18Ai sensitivity observed with sgRNAs targeting Rod, a component required for corona integrity (Raisch et al, 2022) (Fig. 2C). This argues that Cyclin B1-mediated activation of CDK1, but not MAD1 recruitment to the corona, is required for KIF18Ai toxicity.

## Strong reliance on KIF18A for kinetochore–microtubule attachment stability heightens KIF18Ai toxicity

The identification of the minus-end directed kinesin HSET as a KIF18Ai resistance gene led us to speculate that loss of the protein may indirectly silence the SAC and permit mitotic exit by counteracting mitotic spindle defects caused by KIF18A inhibition. HSET cross-links and laterally slides microtubule bundles to control mitotic spindle length (Cai et al, 2009; Steblyanko et al, 2020), and loss of HSET and KIF18A lead to seemingly opposing phenotypes: spindle shortening versus spindle lengthening, respectively. We therefore sought to monitor how the loss of HSET counteracted mitotic phenotypes resulting from KIF18Ai.

In agreement with previous literature, live-cell time-lapse microscopy revealed that KIF18Ai treatment of HCC1806 cells led to a dramatic increase in severely misaligned chromosomes that transiently oscillated far outside the metaphase plate (Figs. 3A and EV4A; Movie EV4). Poor metaphase organization in KIF18Ai-treated cells correlated with a dramatic increase in mitotic duration and errors. Knockout of HSET and, to a lesser extent Cyclin B1, dramatically improved the ability to form and maintain a metaphase plate in KIF18Ai (Fig. 3B). By contrast, MAD1 hypomorphs did not rescue chromosome alignment in KIF18Ai. We conclude that loss of HSET and Cyclin B1 counteract the effect of KIF18Ai treatment in part by dampening chromosome alignment defects.

The dramatic chromosome misalignment phenotype in KIF18Ai led us to speculate that poorly aligned chromosomes play a dominant role in SAC signaling and the resulting mitotic arrest. Fixed immunofluorescence analysis of KIF18Ai-treated HCC1806 cells revealed that, while kinetochores near the spindle pole were more likely to be SAC-active, the majority of SAC signaling

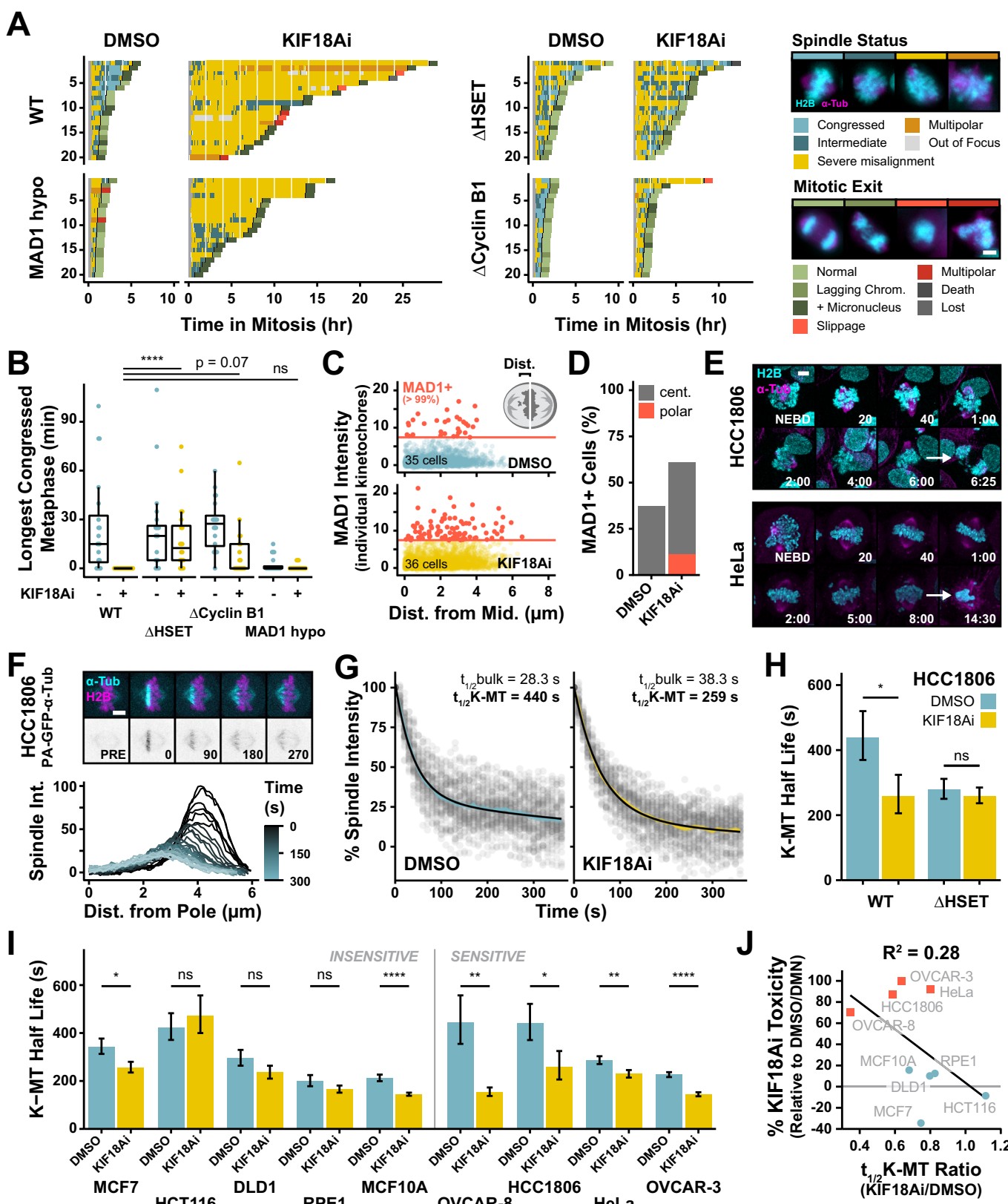

**Figure 3.  KIF18Ai toxicity is relieved by stabilizing kinetochore–microtubule attachments and rescuing metaphase plate congression.**

(A) Quantification of metaphase plate congression and mitotic outcome from live-cell time-lapse wide-field fluorescent microscopy of dividing HCC1806 monoclonal rescue cells in DMSO or KIF18Ai. $n = 20$ cells/condition. Scale bar $= 10\,\mu m$. (B) Longest continuously congressed metaphase from movies in (A). Data are shown as box-and-whisker plots with minimum, first quartile (Q1), median, third quartile (Q3), and maximum. Outliers ($> Q3 + 1.5 \times IQR$) are excluded. $N = 20$ cells per condition. Statistical significance was determined using a one-way-ANOVA with post hoc Dunnett's multiple comparisons test between WT and edited cell lines in KIF18Ai. Sample size and full statistical results are listed in Dataset EV2. (C) Quantification of MAD1 intensity at kinetochores relative to kinetochore position along the mitotic spindle axis in DMSO ($N = 35$ cells, $n = 3104$ kinetochores) or KIF18Ai ($N = 36$ cells, $n = 3369$ kinetochores) treated HCC1806 cells from wide-field immunofluorescence images dataset from Fig. 4C. MAD1+ is defined as >99th percentile signal intensity of DMSO condition. (D) Percent of cells containing at least one MAD1+ kinetochore in either the central ($\leq 5\,\mu m$ from midline) or polar ($> 5\,\mu m$ from midline) regions in DMSO or KIF18Ai conditions from (C). (E) Representative still images from live-cell confocal time-lapse movies of dividing H2B/α-Tubulin fluorescently tagged HCC1806 and HeLa cells in KIF18Ai. Scale bar $= 5\,\mu m$. (F) Top: Representative stills from live-cell confocal time-lapse movies of HCC1806 PA-GFP-α-Tubulin cells. Scale bar $= 5\,\mu m$. Bottom: Intensity distribution over time of photoactivated PA-GFP-α-Tubulin band. (G) Two-phase exponential decay fit of integrated intensity measurements for HCC1806 PA-GFP-α-Tubulin cells in DMSO or KIF18Ai. Shaded colored region represents mean ± SEM. Gray dots represent individual measurements. $N > 50$ cells/condition. (H) Second-order (K-MT) half-life measurements from WT or ΔHSET HCC1806 PA-GFP-α-Tubulin cells in DMSO or KIF18Ai. Data are represented as mean ± SEM. $N > 50$ cells/condition. Statistical significance was determined using an unpaired two-tailed Student's *t* test between DMSO and KIF18Ai treatment for each cell line. Sample size and full statistical results are listed in Dataset EV2. (I) Second-order (K-MT) half-life measurements across the panel of cell lines as in (H). Data are represented as mean ± SEM. $N \geq 30$ cells/condition. Statistical significance was determined using an unpaired two-tailed Student's *t* test between DMSO and KIF18Ai treatment for each cell line. Sample size and full statistical results are listed in Dataset EV2. (J) Linear correlation between 5-day KIF18Ai toxicity and K-MT stability disruption by KIF18Ai across cell line panel. Red squares = sensitive, blue dots = insensitive cell lines. $t_{1/2}$K-MT Ratio $= t_{1/2}$K-MT$_{[KIF18Ai]}/t_{1/2}$K-MT$_{[DMSO]}$. Data information: *$P < 0.05$, **$P < 0.01$, and ****$P < 0.0001$ (B, H, I). Source data are available online for this figure.

---

(MAD1 + ) kinetochores remained within the central chromosome mass (Figs. 3C and EV4B). Furthermore, SAC signaling polar kinetochores were present in fewer than 15% of cells, demonstrating that they are not required for mitotic delays (Fig. 3D). Consistently, HeLa cells treated with KIF18Ai experienced a protracted metaphase arrest with all well-congressed chromosomes (Fig. 3E; Movie EV5). Live-cell visualization of SAC signaling with a BUBR1-EGFP reporter in HeLa cells also revealed that KIF18Ai treatment increased BUBR1 recruitment to congressed kinetochores (Fig. EV4C). We conclude that mitotic delays following KIF18A inhibition are caused by SAC activation at both aligned and misaligned kinetochores.

Changes in kinetochore–microtubule dynamics and attachment stability in response to KIF18Ai treatment could lie upstream of both congression defects and increased SAC signaling (Barisic et al, 2021; Jia et al, 2013). To monitor the stability of kinetochore–microtubule attachments in cells, we used an established live-cell time-lapse imaging assay with cells expressing a photoactivatable-GFP-α-Tubulin (PA-GFP-α-Tub) transgene (Samora and McAinsh, 2011) (Fig. 3F). After photoactivation of a spindle region adjacent to the metaphase plate, a two-phase exponential decay curve was fit to the loss of PA-GFP-α-Tubulin signal over time. This decay curve has two half-lives: the faster half-life reports on general spindle microtubule turnover (bulk), and the slower half-life reports on the turnover of comparatively stable chromosome-attached K-fibers (K-MT) (Zhai et al, 1995). Delays in the slower half-life represent more stable K-MT attachments.

Consistent with the observed increase in SAC activation, we found that KIF18Ai treatment led to a 41% reduction in K-MT stability in metaphase HCC1806 cells (Fig. 3G). This was accompanied by an increase in the rate of microtubule poleward flux, as previously observed in *Drosophila melanogaster* cells (Appendix Fig. S3D) (Buster et al, 2007). Interestingly, HSET knockout cells had a reduced baseline K-MT stability but were resistant to further destabilization induced by treatment with KIF18Ai (7% reduction, Fig. 3H and Appendix Fig. S3A–C). We reasoned therefore that it is the degree of K-MT disruption rather than the absolute stability of these attachments that may likely to predict KIF18Ai-driven mitotic delays and toxicity. To test this

prediction, we introduced or acquired PA-GFP-α-Tub in eight additional cell lines in our panel. While KIF18Ai treatment destabilized K-MT attachments in almost all cell lines, sensitive cell lines were more likely to experience greater K-MT disruption (Fig. 3I; Appendix Fig. S3D,E), and this increased destabilization weakly correlated with KIF18Ai toxicity ($R^2 = 0.28$, Fig. 3J). We conclude that a heightened reliance on KIF18A for K-MT stabilization contributes to, but is not fully predictive of, toxicity in KIF18Ai.

## KIF18Ai-driven SAC signaling occurs in both sensitive and insensitive cell lines

We hypothesized that increased reliance on KIF18A for K-MT attachment stability would translate into stronger SAC signaling after KIF18Ai treatment. We therefore extended our fixed immunofluorescence analysis of SAC signaling with KIF18Ai to the entire panel of sensitive and insensitive cell lines. Surprisingly, both sensitive and insensitive cell lines experienced similar increases in the per-kinetochore BUBR1 signal and the number of MAD1+ kinetochores per cell in response to KIF18Ai (Fig. 4A–D). As a notable exception, sensitive MDA-MB-157 cells clearly experienced greater overall SAC signaling, suggesting that these cells may have a unique underlying reliance on KIF18A for SAC silencing. Ultimately, however, this reveals that the magnitude of SAC signaling at individual kinetochores is a poor overall predictor of KIF18Ai sensitivity (Fig. 4B,D).

Taken together, our data show that treatment with KIF18Ai leads to a weakening of K-MT attachments and an increased probability of activating the SAC at all kinetochores, thereby delaying mitotic exit. Ultimately, the mitotic delay, and by extension KIF18Ai toxicity, can be attenuated by reducing the ability to maintain the mitotic state through SAC disruption (MAD1 hypomorph, Cyclin B1 knockout), reduction of CDK1 activity (Cyclin B1 knockout), or stabilization of K-MT attachments and the metaphase plate (HSET knockout). Finally, we find that the degree of BUBR1 and MAD1 recruitment to kinetochores across cell lines does not correlate well with KIF18Ai sensitivity.

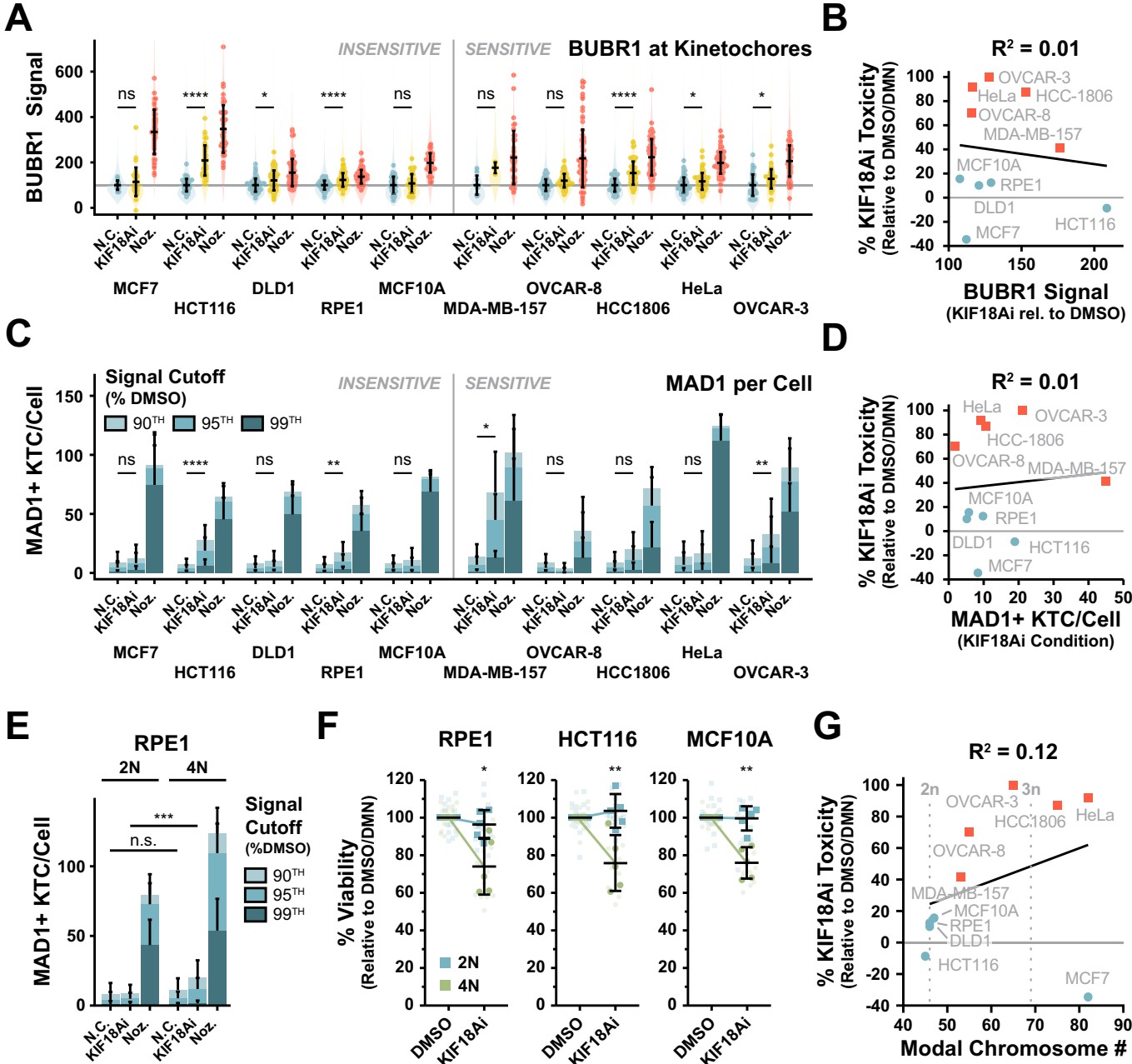

## Hyperploidy increases KIF18Ai SAC signaling to drive toxicity

WGD has been shown to increase sensitivity to KIF18A loss (Quinton et al, 2021), and we considered that higher ploidy might aggravate KIF18Ai toxicity by increasing the number of kinetochores taking part in low-level SAC signaling. To define the impact of chromosome number on KIF18A dependency, we treated matched pairs of diploid and tetraploid RPE1, HCT116, and MCF10A cells with KIF18Ai (Appendix Fig. S4A). Immunofluorescence imaging revealed a modest increase in the total number of MAD1+ kinetochores in KIF18Ai-treated RPE1 tetraploid cells as compared to diploid controls (Fig. 4E). In addition, while the

average per-kinetochore BUBR1 and MAD1 signal was similar or slightly increased between KIF18Ai-treated diploid and tetraploid cells, the cumulative kinetochore signal per cell increased by 56% for BUBR1 and 88% for MAD1 in tetraploid cells treated with KIF18Ai (Appendix Fig. S4B,C). Consistently, doubling chromosome number led to a ~25% increase in toxicity to KIF18Ai in a 5-day viability assay across cell lines (Fig. 4F). However, the KIF18Ai toxicity seen in the tetraploid cells failed to reach the level observed in most sensitive cancer cell lines (Fig. 1C), and chromosome number weakly correlated with KIF18Ai toxicity across the cell line panel ($R^2 = 0.12$, Fig. 4G). We conclude that increases in chromosome number partially sensitize cells to KIF18Ai by increasing the total mitotic SAC burden. However,

**Figure 4.  KIF18Ai-driven SAC signaling occurs in both sensitive and insensitive cell lines and is amplified by whole-genome doubling.**

(A) Intensity of BUBR1 at kinetochores in DMSO, KIF18Ai, and Nocodazole treatments across the full panel of cell lines from wide-field immunofluorescence images. Violin plots summarize all individual kinetochores analyzed, points represent per-cell intensity averages, and error bars are mean ± SD of per-cell averages. $N \geq 4$ cells per condition, $n \geq 730$ kinetochores. Statistical significance was determined using a one-way ANOVA with post hoc Dunnett's multiple comparisons test between drug and DMSO conditions for each cell line. Sample size and full statistical results are listed in Dataset EV2. (B) Linear correlation between 5-day KIF18Ai toxicity and BuBR1 defect across cell line panel. Red squares = sensitive, blue dots = insensitive cell lines. BUBR1 defect = $(Signal_{[KIF18Ai]} - Signal_{[DMSO]})/(Signal_{[Nocodazole]} - Signal_{[DMSO]}) \times 100$. (C) The average number of MAD1+ kinetochores per cell in DMSO, KIF18Ai, and Nocodazole treatments across the full panel of cell lines from wide-field immunofluorescence images. Data are represented as mean ± SD. $N \geq 4$ cells per condition, $n \geq 730$ kinetochores. MAD1+ kinetochores have MAD1 signal >$n$th percentile of signal in DMSO condition as indicated. Statistical significance was determined from >95% MAD1+ kinetochores using a one-way ANOVA with post hoc Dunnett's multiple comparisons test between drug and DMSO conditions for each cell line. Sample size and full statistical results are listed in Dataset EV2. (D) Linear correlation between 5-day KIF18Ai toxicity and number of >95% MAD1+ kinetochores in KIF18Ai across cell line panel. (E) The average number of MAD1+ kinetochores per cell in DMSO, KIF18Ai, and Nocodazole treatments between diploid and WGD RPE1 cell lines from wide-field immunofluorescence images. Data are represented as mean ± SD. $N \geq 25$ cells per condition, $n > 2100$ kinetochores. MAD1+ kinetochores have MAD1 signal >$n$th percentile of signal in DMSO condition as indicated. Statistical significance was determined from >95% MAD1+ kinetochores using an unpaired two-tailed Student's $t$ test for each condition between 2N and 4N cells. Sample size and full statistical results are listed in Dataset EV2. (F) Five-day MTT endpoint viability assay of diploid and WGD cell lines in KIF18Ai. Data are represented as mean ± SD. $N = 5$ independent experiments, $n = 3$ technical replicates per experiment. Statistical significance was determined using an unpaired two-tailed Student's $t$ test. Full statistical results are listed in Dataset EV2. (G) Linear correlation between 5-day KIF18Ai toxicity and modal chromosome number for each cell line taken from the ATCC. Data information: *$P < 0.05$, **$P < 0.01$, ***$P < 0.001$ and ****$P < 0.0001$ (A, C, E, F). Source data are available online for this figure.

this determinant alone is insufficient to explain KIF18A dependency in sensitive cell lines.

## Hyperploidy and low APC/C activity act synergistically to amplify KIF18Ai toxicity

To uncover additional determinants of KIF18A dependency, we ran a genome-wide CRISPR knockout screen in partially sensitive OVCAR-8 cells (70% toxicity; Fig. 1C). We reasoned that the reduced sensitivity of OVCAR-8 cells to KIF18Ai would enable the identification of genes and pathways that increase sensitivity to the drug, in addition to genes promoting resistance. Cas9-expressing OVCAR-8 cells were transduced with a lentiviral knockout library of sgRNAs and selected for 6 days. Cells were then cultured for 4 days in either DMSO or KIF18Ai at $IC_{50}$ and $IC_{90}$ concentrations. Guides that altered growth potential in KIF18Ai conditions relative to control were identified by deep sequencing (Appendix Fig. S5A). We uncovered a similar set of resistance genes to our previous screens, including MAD1, Cyclin B1, HSET, and TRIP13 (Fig. 5A; Appendix Fig. S5B–F; Dataset EV5). By contrast, targeting APC/C subunits or the APC/C E2 ubiquitin ligase UBE2S strongly increased sensitivity to KIF18Ai (Fig. 5A,B). To validate that diminished APC/C activity increased KIF18Ai sensitivity, we used two different sgRNAs to generate polyclonal knockouts of the APC/C subunit APC4 and UBE2S in OVCAR-8 cells (Appendix Fig. S5C). Consistent with the results of the CRISPR screen, KIF18Ai-treated APC4 and UBE2S-edited cell lines experienced a 61% and 86% reduction in viability, respectively, compared to inhibitor-treated WT cells (Fig. 5C,D).

The APC/C is a megadalton-scale E3 ubiquitin ligase responsible for the degradation of Cyclin B1 and Securin to promote the onset of anaphase and timely mitotic exit (Mocciaro and Rape, 2012; Sivakumar and Gorbsky, 2015). Unattached kinetochores generate a SAC signal by catalyzing the assembly of the mitotic checkpoint complex (MCC) that directly binds and inhibits the APC/C to prevent anaphase (Lischetti and Nilsson, 2015). By extension, the metaphase-to-anaphase transition is controlled by a sensitive balance between MCC production and basal APC/C activity (Rieder et al, 1995). We therefore reasoned that KIF18Ai treatment might upset this balance by modestly increasing MCC formation at kinetochores with unstable microtubule attachments. We also

predict that conditions that aggravate a SAC:APC/C imbalance in KIF18Ai, such as hyperploidy or APC/C depletion, would act together to increase KIF18Ai sensitivity. To test this hypothesis, we reduced APC/C activity with shRNAs targeting APC4 or the E2 ligase UBE2S in both diploid (2N) and tetraploid (4N) RPE1 cells (Appendix Fig. S5G). Doubling chromosome number or reduced APC/C activity modestly slowed the growth rate of these insensitive cells treated with KIF18Ai. By contrast, combining low APC/C activity with increased chromosome number almost entirely blocked proliferation in cells treated with KIF18Ai (Fig. 5E,F). We conclude that synergy between hyperploidy-amplified SAC signal and diminished basal APC/C activity is sufficient to generate potent KIF18A dependency.

## KIF18A dependency in cancer relies on a SAC:APC/C imbalance in mitosis

Our data show that disrupting the balance between SAC signaling and APC/C activity is sufficient to generate KIF18A dependency. We therefore set out to test whether such an imbalance exists in sensitive cell lines. The ratio between SAC signaling and basal APC/C activity is difficult to assay directly, but we reasoned that mitotic delays could serve as signatures for increased SAC signaling, while a slowed metaphase-to-anaphase transition would indicate low overall APC/C activity. Indeed, close inspection of earlier live-cell time-lapse experiments revealed KIF18Ai-sensitive cells took more time to complete mitosis in unperturbed conditions (Fig. 6A). Furthermore, mitotic duration proved to be a much stronger correlate for KIF18Ai toxicity ($R^2 = 0.46$) compared to chromosome number ($R^2 = 0.12$). Finally, all sensitive cells and insensitive MCF7 cells displayed an extended metaphase-to-anaphase transition that correlated closely with the propensity of these cell lines to exhibit mitotic delays in KIF18Ai ($R^2 = 0.65$) (Fig. 6B,C).

To examine if the relationship between limited APC/C activity and KIF18Ai toxicity held true across a broad set of cancer cell lines, we interrogated the Broad Dependency Map (DepMap) database which aggregates CRISPR knockout and RNAi whole-genome screening data from over 1000 and 600 cell lines, respectively (Meyers et al, 2017; Tsherniak et al, 2017). The DepMap uses these data to generate gene dependency scores that reflect the fitness change after the disruption of each gene in each

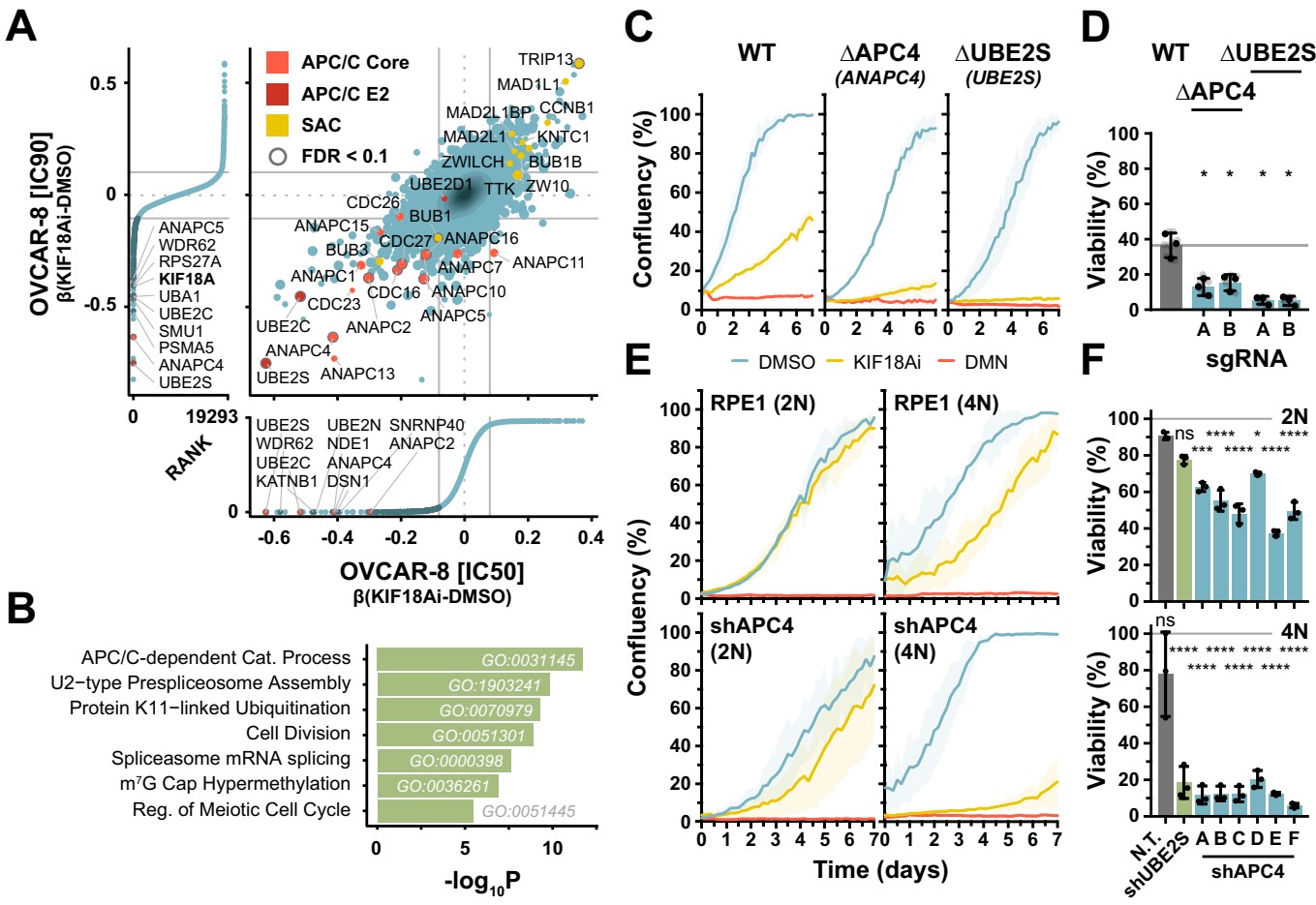

**Figure 5.  Whole-genome doubling synergizes with APC/C defects to induce KIF18A dependency.**

(A) Comparison of OVCAR-8 whole-genome CRISPR–Cas9 screens at IC$_{90}$ and IC$_{50}$ [KIF18Ai]. Data are annotated with mean ± 1.5 × SD. Hits with average FDR < 0.1 are circled in gray in the central plot and highlighted in dark blue in the exterior rank plots. (B) Gene set enrichment analysis of genes from (A) with β(KIF18Ai-DMSO) < 1.5 × SD and average FDR < 0.1. The significance of each GO term was evaluated with a Fisher's exact test using the topGO R package. (C) Longitudinal confluency measurements of OVCAR-8 WT, ΔAPC4 (polyclonal, *sgRNA A*), and ΔUBE2S (polyclonal, *sgRNA A*) in DMSO, KIF18Ai, and Nocodazole conditions. $N = 3$ technical replicates from a single experiment. Data are represented as mean ± SD. (D) Five-day MTT endpoint viability assay of OVCAR-8 polyclonal CRISPR–Cas9 KO cell lines in KIF18Ai. Data are represented as mean ± SD. $N = 3$ independent experiments, $n = 3$ technical replicates per experiment. Statistical significance was determined using a one-way ANOVA with post hoc Dunnett's multiple comparisons test between each gene knockout pair and WT. Full statistical results are listed in Dataset EV2. (E) Longitudinal confluency measurements of shAPC4 and shUBE2S diploid and WGD RPE1 cells in DMSO, KIF18Ai, and Nocodazole conditions. $N = 3$ technical replicates from a single experiment. Data are represented as mean ± SD. (F) 5-day MTT endpoint viability assay of shAPC4 and shUBE2S diploid and WGD RPE1 cell lines in KIF18Ai. Data are represented as mean ± SD. $N = 3$ technical replicates from a single experiment. Statistical significance was determined using a one-way ANOVA with post hoc Dunnett's multiple comparisons test between each 2N and 4N knockdown and 2 N non-targeting sgRNA cell lines. Full statistical results are listed in Dataset EV2. Data information: *$P < 0.05$, ***$P < 0.001$, and ****$P < 0.0001$ (D, F). Source data are available online for this figure.

cell line. Co-dependent genes exhibit similar patterns and magnitudes of fitness change across the cell line panel, and linear regression between cell-specific gene dependency scores is used to assess this relationship. Importantly, using the RNAi dataset which has been shown to be more sensitive for interrogating the effect of depleting essential genes (Krill-Burger et al, 2023), the top three positively correlated co-dependencies for KIF18A's RNAi were the APC/C subunits APC1, APC4, and APC8 (Figs. 6D and EV5A). This genetic relationship suggests that cell lines sensitive to KIF18A depletion are also intolerant of reductions in APC/C activity, likely reflecting a SAC:APC/C imbalance. We therefore conclude that KIF18Ai toxicity and a SAC:APC/C imbalance are broadly linked.

To directly assay APC/C activity in cells, we sought to monitor degradation of the APC/C substrate Cyclin B1 at the metaphase-to-anaphase transition, and we successfully acquired or generated endogenous fluorescent tagging of Cyclin B1 three insensitive (HCT116, DLD1, RPE1) and three sensitive (OVCAR-8, HCC1806, HeLa) cell lines from the full panel. Live-cell experiments showed that endogenously tagged Cyclin B1 was degraded more slowly in sensitive cell lines than insensitive cells in untreated conditions (Fig. 6E–G). Depleting SAC function with the MSP1 inhibitor Reversine largely corrected the degradation delay in OVCAR-8 and HCC1806 but not HeLa cells (Fig. EV5B,C), suggesting that weakened APC/C activity at the metaphase-to-anaphase transition can be caused by a combination of both weak basal APC/C activity (HeLa cells) and/or an underlying

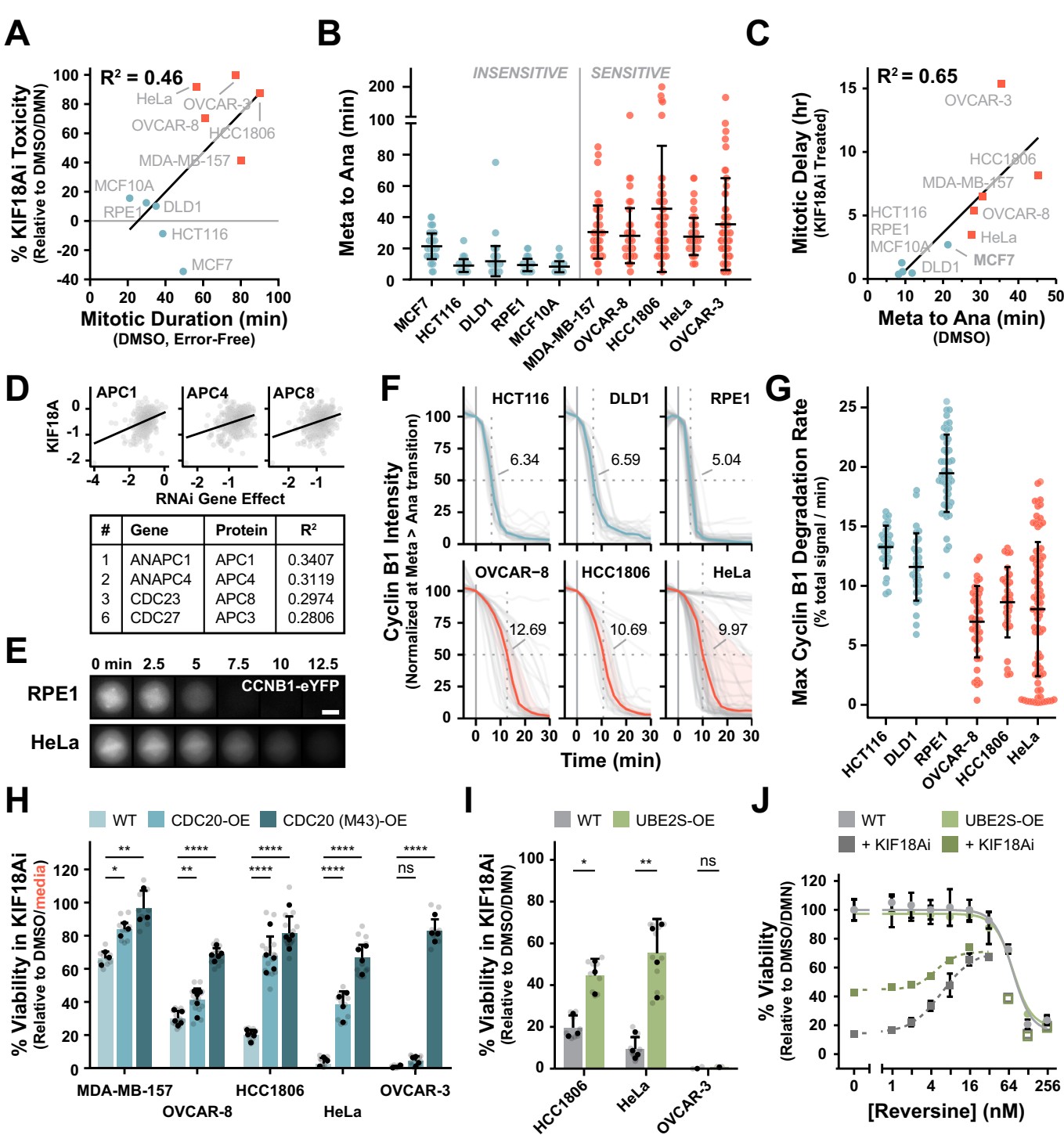

defect in SAC silencing (OVCAR-8 and HCC1806 cells). Furthermore, while Cyclin B1 degradation was modestly delayed in insensitive cells treated with KIF18Ai, Cyclin B1 destruction rates were dramatically suppressed in sensitive cell lines, consistent with the chronic SAC activation leading to delayed mitotic progression. Taken together, these data provide evidence for an underlying SAC:APC/C imbalance in sensitive OVCAR-8, HCC1806, and HeLa cells, which primes these cells for mitotic delay in KIF18Ai.

## Increasing APC/C activity rescues KIF18Ai toxicity in sensitive cell lines

Our data predict that increasing APC/C activity would correct the SAC:APC/C imbalance and reduce KIF18Ai toxicity in sensitive cell lines. Work in yeast has shown that high CDC20 expression levels increase the ability to activate the APC/C and exit mitosis independently of the SAC (Bonaiuti et al, 2018). Furthermore, an

◀

**Figure 6. Low basal APC/C activity is a hallmark of KIF18A-dependent cell lines.**

(A) Linear correlation between 5-day KIF18Ai toxicity and mitotic duration in cells without errors from live-cell wide-field time-lapse microscopy of H2B/α-Tubulin fluorescently tagged cell lines from Fig. 1D. (B) Metaphase-to- anaphase duration from live-cell wide-field time-lapse microscopy of H2B/α-Tubulin fluorescently tagged cell lines from Fig. 1D. Error bars represent mean ± SD. $N \geq 50$ cells per condition. Full sample size information is listed in Dataset EV2. (C) Linear correlation between metaphase-to-anaphase duration in (B) and mitotic duration in KIF18Ai from live-cell wide-field time-lapse microscopy of H2B/α-Tubulin fluorescently tagged cell lines from Fig. 1D. (D) Top KIF18A co-dependency relationships from the DepMap RNAi dataset. $N > 600$ cell lines. (E) Live-cell wide-field time-lapse microscopy of endogenously tagged Cyclin B1-eYFP at the metaphase-to- anaphase transition for RPE1 and HeLa cells. Scale bar = 10 μm. (F) Quantification of Cyclin B1 degradation rates as in (E) for untreated endogenously tagged fluorescent cells. The median (colored line) of individual traces (gray lines) is plotted, with the shaded region encompassing the first and third quartile of each population. Cyclin B1 signal is normalized to the metaphase inflection point, and median $t_{1/2}$ values are listed. Full sample size information is listed in Dataset EV2. (G) Maximum slope of Cyclin B1 degradation at metaphase for untreated endogenously tagged fluorescent cells in (F). Rates are calculated relative to the total Cyclin B1 signal at mitotic entry. Data are represented as mean ± SD. $N \geq 30$ cells per condition. (H) Media normalized 5-day MTT endpoint viability assay of sensitive cell lines overexpressing either WT or M43 CDC20 in KIF18Ai. Data are represented as mean ± SD. $N \geq 3$ independent experiments per cell line, $n = 3$ technical replicates per experiment. Media normalization was used rather than DMN normalization since CDC20 overexpression generated growth rescue in DMN. Statistical significance was determined using a one-way ANOVA with post hoc Dunnett's multiple comparisons test between each overexpression pair and WT. Full statistical results are listed in Dataset EV2. (I) Five-day MTT endpoint viability assay of UBE2S-overexpressing cell lines in KIF18Ai. Data are represented as mean ± SD. $N \geq 1$ independent experiment, $n = 3$ technical replicates per experiment. Statistical significance was determined using an unpaired two-tailed Student's $t$ test between WT and UBE2S-overexpressing cells. Full statistical results are listed in Dataset EV2. (J) Titration of the MPS1 inhibitor Reversine in a 5-day MTT endpoint viability assay in DMSO or KIF18Ai-treated HCC1806 WT or UBE2S-overexpressing cells. Open squares are omitted from the curve fit. $N = 3$ technical replicates from a single experiment. Data are represented as mean ± SD. Data information: *$P < 0.05$, **$P < 0.01$, and ****$P < 0.0001$ (H, I). Source data are available online for this figure.

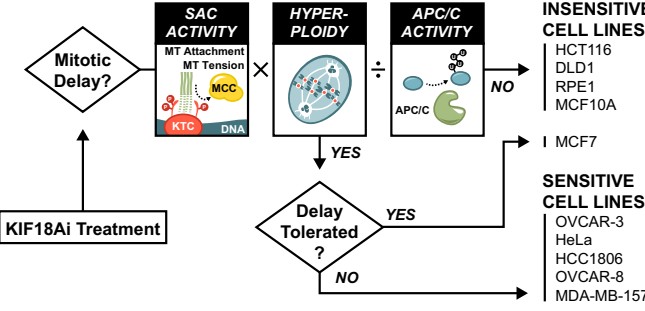

**Figure 7. A model for KIF18A dependency.**

Toxicity in KIF18Ai is the result of mitotic delays and errors. Whether a cell arrests in mitosis depends on three factors (i) the amount of elevated SAC activity at each kinetochore multiplied by (ii) the number of kinetochores and mitigated by (iii) the basal activity of the APC/C. Cells with mitotic delays resulting from these factors are generally sensitive to KIF18Ai. However, toxicity can be rescued by hyperstability of the mitotic spindle apparatus as it is with MCF7 cells.

alternative CDC20 (M43) translational isoform lacking the first 42 amino acids was shown to activate the APC/C and promote mitotic exit in a manner that was resistant to SAC production (Tsang and Cheeseman, 2023). We therefore predicted that overexpression of either WT or the M43 isoform of CDC20 would alleviate KIF18Ai toxicity by increasing overall APC/C activity relative to the SAC. Indeed, overexpression of WT CDC20 and especially the M43 translational isoform provided a robust increase in cell viability in response to KIF18Ai treatment in all the sensitive cells (Fig. 6H; Appendix Fig. S6A). Consistent with previous work, these cell lines also experienced increased growth in DMN (Appendix Fig. S6B). Taken together, these data show that genetically tuning the balance of APC/C activity relative to the SAC is sufficient to reduce KIF18Ai sensitivity.

We next set out to test if increasing basal APC/C activity by overexpression of its E2 ligase UBE2S (Craney et al, 2016; Garnett et al, 2009) would be sufficient to reduce KIF18Ai toxicity. Indeed, overexpression of UBE2S resulted in a partial reduction in KIF18Ai toxicity in HCC1806 and HeLa cells, but not OVCAR-3 cells

(Fig. 6I; Appendix Fig. S6C). This suggests that either UBE2S is not rate-limiting for APC/C activity in OVCAR-3 cells or that APC/C hyperactivity is insufficient to overcome the KIF18Ai-generated SAC signaling in these cells. Nevertheless, we conclude that heightened basal APC/C activity can be protective against KIF18A inhibition.

Finally, to examine whether heightened APC/C activity could directly increase the tolerance for SAC signaling in KIF18Ai, we titrated Reversine into WT and UBE2S-overexpressing HCC1806 cells treated with KIF18Ai. As before, partial SAC inhibition with low doses of Reversine led to a potent rescue of KIF18Ai toxicity, while higher concentrations were toxic to cells (Fig. 6J). The maximal viability rescue was not additive between UBE2S over-expression and Reversine, suggesting that these rescues act through the same pathway. However, maximal rescue was reached at lower concentrations of Reversine in UBE2S-overexpressing cells, indi-cating that increased APC/C activity directly translates into a higher tolerance for the low-level SAC signaling caused by KIF18Ai. In other words, high basal APC/C activity can directly mitigate low levels of SAC signaling to promote anaphase onset.

## Discussion

Identifying appropriate patient populations is vital to the clinical success of KIF18A inhibitors. Previous work has identified WGD and CIN as indicators of KIF18A dependency (Marquis et al, 2021; Quinton et al, 2021). However, these correlates were only weakly predictive of KIF18Ai sensitivity, suggesting that additional determinants remained to be uncovered. Furthermore, the Broad DepMap cancer database reports that KIF18A dependency is not readily predicted by expression, lineage, or mutational datasets (Dempster et al, 2020), reinforcing the need for a mechanistic understanding of KIF18A dependency. Here, we show that toxicity from KIF18A inhibition is caused by prolonged mitotic delays mediated by persistent SAC signaling. Treatment with KIF18Ai generates a minor increase in SAC signaling across all cell lines due to a weakening of microtubule-kinetochores attachments. Whether cell lines tolerate this defect and enter anaphase depends on the

magnitude of this ploidy-amplified SAC burden as well as the preexisting balance between the SAC and APC/C at mitotic exit. Taken together, we show that increased SAC signal on top of a functionally weakened APC/C at metaphase plays a central role in driving KIF18Ai sensitivity (Fig. 7).

Several lines of evidence underscore the central role of SAC signaling and APC/C activity in mediating KIF18Ai-driven mitotic delays and toxicity. First, decreasing APC/C activity or increasing ploidy produces modest increases in KIF18A sensitivity that synergize strongly. Second, there is a strong correlation between the duration of the metaphase-to-anaphase transition and KIF18Ai toxicity. Third, we show that a reduction of SAC signaling, through genetic or pharmacological means, alleviates KIF18Ai toxicity and may present opportunities for resistance. In this vein, others have shown that knockout of the SAC component MAD2 increases the viability of KIF18A-depleted cells (Janssen et al, 2018). Finally, increases in basal APC/C activity provide resistance to KIF18Ai. Thus, increased SAC signal or decreased APC/C activity predispose cells to metaphase delays and KIF18Ai sensitivity.

A unique feature of KIF18A inhibition compared with other antimitotic agents is its ability to apply a gentle pressure toward mitotic delay. Unlike the microtubule-stabilizing agent Taxol, which shows broad cytotoxicity by promoting persistent SAC signaling, KIF18A inhibition generates a small increase in SAC signal through unstable kinetochore–microtubule attachments. In doing so, KIF18Ai selectively targets cells that are already predisposed to mitotic delays while sparing cells that progress through mitosis quickly. Indeed, the best predictor of KIF18Ai sensitivity is the duration of unperturbed mitosis. Consistently, the intermediate increase in SAC signaling with KIF18Ai generates a graded severity of mitotic responses, as with the increased sensitivity observed with hyperploid cell lines that magnify the total SAC burden.

A better appreciation of the role of the APC/C in generating KIF18Ai sensitivity raises the question of why cancer cells would select for APC/C dysfunction. A recent report showed that lowered APC/C activity plays a protective role by tempering excessive CIN (Sansregret et al, 2017). In these cells, the benefits of tumor heterogeneity are balanced against the fitness cost of genome instability to maximize tumor growth potential. This may help explain why CIN cancers have been shown to be, on average, more sensitive to KIF18A loss (Marquis et al, 2021). In addition, reduced APC/C activity is a known resistance mechanism to SAC-targeting antimitotic agents such as MPS1 inhibitors (Thu et al, 2018). Typically, SAC ablation results in premature and catastrophic mitotic exit, however, lowered APC/C activity slows the rate of Cyclin B1 and Securin degradation, granting additional time for correct chromosome alignment before anaphase onset. We therefore speculate that KIF18A inhibitors might serve as a second-line treatment for tumors that have developed resistance to SAC inhibitors.

Although our data provide functional evidence for a SAC:APC/C imbalance in KIF18Ai-sensitive cell lines, there are many ways by which this imbalance can be achieved. For instance, SAC signaling strength can be altered by ploidy, baseline K-MT attachment quality, a heightened reliance on KIF18A for K-MT attachment stabilization, the abundance and recruitment of SAC signaling proteins, the abundance and recruitment of SAC-silencing phosphatases, or the efficiency of MCC recycling factors. Similarly,

APC/C function may be limited by the expression of any one of its subunits, abundance and activity of E2 conjugating enzymes, or expression and recruitment of its mitotic coactivator CDC20. In addition, expression levels of CDC20 translational isoforms make the APC/C more or less sensitive to SAC inhibition (Tsang and Cheeseman, 2023). In short, the combination of alterations that generate the same phenotypic outcome could differ from one cancer to the next. Perhaps because of this, we have been unable to identify specific mutations or expression pattern changes that can predict KIF18A dependency with high confidence. On the contrary, we show that basal mitotic duration correlates well with KIF18Ai toxicity because it wholistically reports on the SAC:APC/C imbalance in a way that is agnostic of the underlying dysfunction. However, although mitotic duration can be established experimentally in cell lines, it cannot be directly assayed in patient tumors. This highlights an opportunity to identify better biomarkers that can accurately report on this imbalance, and by extension, KIF18A dependency.

Potent KIF18Ai toxicity is generated when cells suffer a protracted mitotic delay and lethal errors in cell division. However, a surprising feature of KIF18Ai toxicity was the variation in response to mitotic delay. For example, while HeLa cells treated with KIF18Ai had highly stable metaphase plates that eventually underwent cohesion fatigue, HCC1806 cells struggled to form and maintain a congressed metaphase. By contrast, insensitive MCF7 cells had robust mitotic spindles that could delay in mitosis up to seven hours and ultimately divide normally. This suggests that spindle robustness plays an important role in determining sensitivity to KIF18Ai and possibly other drugs that induce mitotic delays. Understanding this cellular response to mitotic delays is an underappreciated feature of antimitotic therapies and an opportunity for further research.

## Methods

No statistical methods were used to predetermine the sample size. The experiments were not randomized, and investigators were not blinded to allocation during experiments and outcome assessment.

### Cell lines and culture conditions

HeLa, HCT116, and DLD1 cells were grown in DMEM medium (Corning Cellgro) containing 10% fetal bovine serum (Sigma), 100 U/mL penicillin, 100 U/mL streptomycin and 2 mM L-glutamine. hTERT RPE1 cells were grown in DMEM:F12 medium (Corning Cellgro) containing 10% fetal bovine serum (Sigma), 0.348% sodium bicarbonate, 100 U/mL penicillin, 100 U/mL streptomycin and 2 mM L-glutamine. MDA-MB-157, OVCAR-8, HCC1806, and MCF7 cells were grown in RPMI 1640 medium (Thermo Fisher Scientific) containing 10% fetal bovine serum (Sigma), 100 U/mL penicillin, 100 U/mL streptomycin and 2 mM L-glutamine. OVCAR-3 cells were grown in RPMI 1640 medium (Thermo Fisher Scientific) containing 20% fetal bovine serum (Sigma), 100 U/mL penicillin, 100 U/mL streptomycin, 2 mM L-glutamine, and 10 µg/ml bovine insulin (Sigma). MCF10A cells were grown in DMEM:F12 medium (Corning Cellgro) containing 5% horse serum (Invitrogen), 100 U/ml penicillin, 100 U/ml streptomycin, 20 ng/mL hEGF, 0.5 µg/mL hydrocortisone (Sigma),

50 ng/mL cholera toxin (Sigma) and 10 µg/ml bovine insulin (Sigma). All cell lines were maintained at 37 °C in a 5% $CO_2$ atmosphere with 21% oxygen. STR profiling was performed to verify cell line identity and cultures were routinely checked for mycoplasma contamination.

HeLa MAD1 knockout FRT TetON VSV-MAD1(WT), HeLa MAD1 knockout FRT TetON VSV-MAD1(E52K, E53K, E56K), and RPE1 Cyclin B1-EYFP cells were a kind gift of A. Saurin (Uni. If Dundee). RPE1 PA-GFP-α-Tubulin cells were a kind gift of D. Compton (Dartmouth College). Paired genome doubled cell lines RPE1 (2N) and (4N), HCT116 (2N) and (4N), and MCF10A (2N) and (4N) cells were a kind gift of N. Ganem (Boston University).

## Gene targeting and stable cell lines

To generate fluorescent histone H2B and α-Tubulin-labeled cell lines, ORFs were cloned into FUGW lentiviral vectors. Fluorescent populations of cells were then generated by lentivirus-mediated transduction. DLD1 cells were transduced with H2B-mRFP and YFP-α-Tubulin. HCT116 and MCF10A cells were transduced with H2B-iRFP. HCC1806, MCF7, MDA-MB-157, RPE1, and OVCAR-8 cells were transduced with H2B-iRFP and eGFP-α-Tubulin. HeLa cells were transduced with H2B-mRFP and eGFP-α-Tubulin. OVCAR-3 cells were transduced with H2B-eGFP. Polyclonal populations of cells expressing the desired fluorescent markers were used directly or isolated using FACS.

To generate CRISPR–Cas9-mediated knockout lines, sgRNAs targeting APC4 (ANAPC4-A, 5′-aacatgtatgtgtgaagcat-3′; ANAPC4-B, 5′-gtcacagaagtctctaccaa-3′), Cyclin B1 (CCNB1-A, 5′-gtcagaccaaaatacctact-3′; CCNB1-B, 5′-gaggccaagaacagctcttg-3′), EB1 (MAPRE1-A, 5′-tggaaaagactatgaccctg-3′; MAPRE1-B, 5′-ctcaacacagagaaccgctg-3′), HSET (KIFC1-A, 5′-actggaggggcatttagcca-3′; KIFC1-B, 5′-gcatactggatagccatcca-3′), MAD1 (MAD1L1-A, 5′-gaagaagcgcgagacccacg-3′; MAD1L1-B, 5′-gctggacctgcaacacaagt-3′), Rod (KNTC1-A, 5′-aagctaacgatgaaaatcgg-3′; KNTC1-B, 5′-aaacattcggaacactatgg-3′), TRIP13 (TRIP13-A, 5′-cgagtcgccaacggtccacg-3′; TRIP13-B, 5′-ttgtgtttggtgattacaca-3′), and UBE2S (UBE2S-A, 5′-aactcaccagcagtacgtgt-3′; UBE2S-B, 5′-catcaaggtctttcccaacg-3′) were cloned into either LentiCRISPR v2 puro or LentiCRISPR v2 blast vectors (# 52961, #83480; Addgene). Knockout cells were then generated by lentivirus-mediated transduction of these constructs. Positive selection of transduced cells was performed 2 days after transfection with 1 µg/mL puromycin or 5 µg/mL blasticidin, respectively. Monoclonal cell lines were isolated by limiting dilution. The ablation of protein production was assessed by immunoblotting.

To generate the HCC1806 H2B-iRFP photoactivatable (PA)-GFP-α-Tubulin WT and ΔHSET cell lines, H2B-iRFP was cloned into a FUGW lentiviral vector and introduced to cells by lentivirus-mediated transduction. Fluorescent cells were isolated using FACS. PA-GFP-α-Tubulin was then cloned into a modified FUGW lentiviral vector with a puromycin-resistance cassette and introduced to cells by lentivirus-mediated transduction. Monoclonal cell lines were isolated by limiting dilution, and PA-GFP-α-Tubulin expression was evaluated using fluorescence microscopy.

To generate the panel of H2B-mRuby3 photoactivatable (PA)-GFP-α-Tubulin cell lines, A modified FUGW lentiviral vector expressing PA-GFP-α-Tubulin-T2A-H2B-mRuby3 with a puromycin-resistance cassette was introduced to cells by lentivirus-mediated transduction. Positive selection of transduced cells was performed 2 days after transfection with 1 µg/mL puromycin. Fluorescent cells were then isolated using FACS.

To generate overexpression HCC1806 KIF18A mutant cell lines, the following ORFs were cloned into a modified FUGW lentiviral vector with a puromycin-resistance cassette: KIF18A-(3x)HA, WT; KIF18A(G289I)-(3x)HA, drug-resistant; KIF18A(G289I, R308A, K311A)-(3x)HA, drug-resistant motor dead; KIF18A(G289I, V614A, W617A)-(3x)HA, drug-resistant PP1-binding mutant. Overexpression cells were then generated by lentivirus-mediated transduction of these constructs. Positive selection of transduced cells was performed 2 days after transfection with 1 µg/mL puromycin. Expression levels of the transgenes were assessed by immunoblotting.

To generate HeLa endogenously tagged CCNB1-EYFP (Cyclin B1) cells, an sgRNA targeting the Cyclin B1 translational stop codon (5′-gtgtaacttgtaaacttgagt-3′) was cloned into a pX459 vector (#62988; Addgene). A plasmid vector containing >950 bp gene homology arms and an EYFP tag was used as a repair template (kind gift of A. Saurin). HeLa cells were transiently transfected (X-tremeGENE HP, Roche) with the pX459 plasmid and repair vector. Fluorescent cells were isolated 6 days after transfection by FACS. Due to low efficiency, a second round of FACS was run to further isolate fluorescent cells 2 weeks later.

To generate HCT116, DLD1, OVCAR-8, and HCC1806 endogenously tagged CCNB1-mNeonGreen (Cyclin B1) cells, an Alt-R™ crRNA targeting the Cyclin B1 translational stop codon (5′-AltR1-ucaagauuuagccaaggcugguuuuagagcuaugcu-AltR2-3′; IDT) was annealed with a tracrRNA (#1072533; IDT) and combined with recombinant S.p. Cas9 nuclease (#1081059; IDT). The prepared RNP was nucleofected into cells using the Lonza 4D-Nucleofector system with the SE reagent and following protocols. HCT116 – EN-113. DLD1 & OVCAR-8 – EN-130. HCC1806 – FF-150. mNeonGreen-expressing cells were isolated after a week by two rounds of FACS sorting.

To generate HCC1806 H2B-iFP, EGFP-α-Tubulin, UBE2S overexpression cell lines, the UBE2S ORF was cloned into a modified FUGW lentiviral vector with a puromycin-resistance cassette and introduced to H2B- and -α-Tubulin-tagged cells by lentivirus-mediated transduction. Positive selection of transduced cells was performed 2 days after transfection with 1 µg/mL puromycin. UBE2S expression levels were assessed by immunoblotting.

## RNA interference

Dharmacon pGIPZ lentiviral vectors containing shRNAs targeting UBE2S (5′-acaaatccaggtcccagtg-3′), or APC4 (ANAPC4-A, 5′-tatctctggagctaaagcg-3′; ANAPC4-B, 5′-tatgagtaaactttctggc-3′; ANAPC4-C, 5′-agtccatctcctatgtcct-3′; ANAPC4-D, 5′-aactgattcatcaagagag-3′; ANAPC4-E, 5′-tacaatggaatacagattg-3′; ANAPC4-F, 5′-tttcctgcacaaacttggt-3′) were purchased (Horizon). Stable shRNA-mediated knockdown cell lines were generated by lentivirus-mediated transduction. Positive selection of transduced cells was performed 2 days after transfection with 1 µg/mL puromycin. Knockdown efficiency was assessed by immunoblotting.

## Lentiviral production and transduction

Lentiviral expression vectors were co-transfected into HEK293FT cells with the lentiviral packaging plasmids psPAX2 and pMD2.G (#12260 and #12259; Addgene). In brief, $3 \times 10^6$ HEK293FT cells were seeded into a poly-L-lysine-coated 10-cm culture dish the day before transfection. For each 10-cm dish, the following DNA was diluted in 0.6 ml of OptiMEM (Thermo Fisher Scientific): 4.5 µg of lentiviral vector, 6 µg of psPAX2 and 1.5 µg of pMD2.G. Separately, 35 µL of 1 µg/µL 25 kDa polyethylenimine (PEI; Sigma) was diluted into 600 µL of OptiMEM, briefly vortexed and incubated at room temperature for 5 min. After incubation, the DNA and PEI mixtures were combined, briefly vortexed and incubated at room temperature for 20 min. During this incubation, the culture medium was replaced with 8 ml of pre-warmed DMEM + 1% FBS. The transfection mixture was then added dropwise to the 10-cm dish. Viral particles were collected 48 h after the medium change and filtered through a 0.45-µm PVDF syringe filter. The filtered supernatant was used directly to infect cells. Aliquots were snap-frozen and stored at −80 °C. For transduction, lentiviral particles were diluted in complete growth medium supplemented with 10 µg/ml polybrene (Sigma) and added to cells.

## Chemical inhibitors

AM-1882 (KIF18Ai; Amgen) was dissolved in DMSO and used at a final concentration of 250 nM or 500 nM, unless otherwise indicated. Reversine (MPS1i; Axon Med Chem) was dissolved in DMSO and used at a final concentration of 500 nM, unless otherwise indicated. RO-3306 (Sigma) was dissolved in DMSO and used at a final concentration of 10 µM, unless otherwise indicated. Taxol (Sigma) was dissolved in DMSO and used at a final concentration of 10 µM. MG132 (Sigma) was dissolved in DMSO and used at a final concentration of 10 µM. Nocodazole (Sigma) was dissolved in DMSO and used at a final concentration of 3.3 µM. Dimethylenastron (DMN; Sigma) was dissolved in DMSO and used at a final concentration of 2 µM. (-)-Nutlin-3 (Cayman Chemical Company) was dissolved in DMSO and used at a final concentration of 5 µM.

## Antibody techniques

For immunoblot analyses, protein samples were separated by SDS–PAGE, transferred onto nitrocellulose membranes with a Trans-Blot Turbo Transfer System (BioRad) and then probed with the following primary antibodies: α-Tubulin (rat; Thermo Fischer, YL1/2, #MA1-80017; 1:1000–1:3000), APC4 (rabbit, Bethyl Laboratories, #A301-176A; 1:1000), Cyclin B1 (mouse; Santa Cruz, GNS1, #SC245; 1:750-1:1000), HA (rat; Roche, #ROAHAHA; 1:1000), HSET (rabbit; Cell Signaling, #12313; 1:1000), KIF18A (rabbit; Bethyl, #A301-080; 1:1000), MAD1 (mouse; Millipore Sigma, #MABE867; 1:1000), TRIP13 (mouse; Santa Cruz, A-7, #SC514285; 1:1000), UBE2S (rabbit; Proteintech, #14115-1AP; 1:1000), Vinculin (mouse; Santa Cruz, 7F9, #SC73614; 1:1000). Proteins were then detected using HRP-conjugated anti-mouse (horse; Cell Signaling, #7076; 1:1000), HRP-conjugated anti-rat (goat; Cell Signaling, #7077; 1:1000), or HRP-conjugated anti-rabbit (goat; Cell Signaling, #7074; 1:1000) and SuperSignal West

chemiluminescence substrate (Pico PLUS/Femto, Thermo Fischer). Signals were visualized and acquired using a Genesys G:Box Chemi-XX6 system (Syngene).

For immunofluorescence, cells were grown on 12-mm glass coverslips. If mitotic cells were being analyzed, they were washed for 30 s in microtubule-stabilizing buffer (MTSB; 1 mM EGTA, 1 mM $MgSO_4$, 100 mM PIPES, 30% glycerol) before fixation in 4% formaldehyde at room temperature for 10 min. Otherwise, cells were directly fixed in MeOH at −20 °C for 10 min. Cells were blocked in 2.5% FBS, 200 mM glycine, and 0.1% Triton X-100 in PBS for 1 h. Antibody incubations were conducted in the blocking solution for 1 h. DNA was stained with DAPI and cells were mounted in ProLong Gold Antifade (Invitrogen). Staining was performed with the following primary antibodies: α-Tubulin (rat; Thermo Fischer, YL1/2, #MA1-80017; 1:1000), BUBR1 (sheep; kind gift of S. Taylor (Univ, of Manchester) #SBR1.1; 1:1000), CENP-A (mouse; GeneTex, #GTX13939; 1:2000), CEP192–Cy5 (directly labeled goat; raised against CEP192 amino acids 1–211, this study; 1:1000), KIF18A (rabbit; Bethyl, #A301-080; 1:1000), MAD1 (mouse; Millipore Sigma, #MABE867; 1:1000), Phospho-Histone H2A.X (Ser139) (rabbit; Cell Signaling, # 2577; 1:800). Immunofluorescence images were collected using a Deltavision Elite system (GE Healthcare) controlling a Scientific CMOS camera (pco.edge 5.5). Acquisition parameters were controlled by Soft-WoRx suite (GE Healthcare). Images were collected at room temperature (25 °C) using an Olympus $40 \times 1.4$ NA oil objective or an Olympus $100 \times 1.4$ NA oil objective at 0.2-µm z-sections. Images were acquired using Applied Precision immersion oil ($N = 1.516$).

Quantification of location and signal intensity at individual kinetochores was achieved using a custom FIJI macro on deconvolved 3D image z-stacks. Briefly, individual kinetochore regions were defined using tools from the 3D ImageJ Suite plugin (Ollion et al, 2013) on manually thresholded BUBR1 image stacks. 3D-segmented kinetochore regions were then used to measure BUBR1 and MAD1 intensity. Background signal was determined as the median signal of the non-kinetochore area throughout the entire z-stack and subtracted from BUBR1 and MAD1 measurements, respectively. The location of spindle poles was identified using a similar methodology using the CEP192 channel. Downstream processing and quality control of kinetochore intensity data was managed in R. Non- or missegmented-kinetochores were excluded based on size to remove instances where multiple kinetochores segmented together ($> 1.5 \times$ interquartile range (IQR) of average volume) or by coincidence of CEP192 signal (spindle poles). Cells with aberrant kinetochore numbers (outside $1.5 \times$ IQR of average kinetochore number) were also excluded from the analysis. MAD1+ kinetochores were determined as being above the 90th, 95th, or 99th percentile of MAD1 signal in the DMSO condition as indicated.

## Live-cell microscopy

Fluorescent cell lines were seeded into either 4-chamber, 35-mm glass-bottom culture dishes (Greiner), four-well chamber slides (Ibidi), or eight-well chamber slides (Ibidi). The day of the experiment, cells were transferred to $CO_2$-independent base medium (Thermo Fischer) with the appropriate additives for each cell line and maintained at 37 °C in an environmental control

station. Long-term time-lapse imaging was performed using a Deltavision Elite system (GE Healthcare) controlling a Scientific CMOS camera (pco.edge 5.5.). Images were acquired with an Olympus $40 \times 1.4$ NA oil objective. For general mitotic phenotypes, cells were treated as indicated, then images were captured every 5 min in $9 \times 3\,\mu m$ z-sections in respective fluorescent channels and by differential inference contrast. Phenotypic evaluation and intensity measurements were taken from maximum intensity projected 2D time-lapse images. Mitotic duration was calculated as the time taken from nuclear envelope breakdown to the onset of anaphase. Metaphase-to-anaphase duration was calculated as the time from the appearance of the last, uninterrupted metaphase plate to the onset of anaphase. The degree of chromosome misalignment during mitosis was subjectively evaluated based on the position of the poleward-most chromosomes relative to spindle poles. These observations were standardized using a set of pre-determined reference images.

For Cyclin B1-endogenously tagged experiments, fluorescent cell lines were seeded into either 4-chamber, 35-mm glass-bottom culture dishes (Greiner), four-well chamber slides (Ibidi), or eight-well chamber slides (Ibidi). Time-lapse imaging was performed using a Zeiss Axio Observer 7 inverted microscope with Slidebook 2023 software (3i—Intelligent, Imaging Innovations, Inc.), X-Cite NOVEM-L LED laser and filter cubes, and Prime 95B CMOS camera (Teledyne Photometrics) with a 20× plan-apochromat oil immersion objective with 0.8 NA. During imaging cells were maintained in a stage top incubator with temperature, $CO_2$, and humidity control (3i). Cells were treated as indicated, then images were captured every 2.5 min in $9 \times 3\,\mu m$ z-sections in respective fluorescent channels and by phase contrast. Phenotypic evaluation and intensity measurements were taken from maximum intensity projected 2D time-lapse images in FIJI. Background-subtracted Cyclin B1 intensity was normalized to 100% at anaphase onset for the measurement of half-life values. Background-subtracted maximum mitotic Cyclin B1 intensity was normalized to 100% for degradation rate measurements.

For high-resolution live-cell imaging of mitosis, cells were seeded into 4-chamber, 35-mm glass-bottom culture dishes (Greiner) and maintained at 37 °C and 5% $CO_2$ in an environmental control station. Imaging was performed using a Lecia SP-8 confocal microscope, equipped with a resonance scanner, and 405-nm, 488-nm, 552-nm, and 638-nm laser lines. Images were acquired with a Leica $40 \times 1.3$ NA oil objective. For general mitotic phenotypes, cells were treated as indicated, then images were captured every 5 min in $8 \times 3\,\mu m$ z-sections in respective fluorescent channels and by differential inference contrast. For HeLa EGFP-BUBR1 movies, cells were simultaneously treated with the indicated drugs and MG132 to prevent anaphase onset in all conditions, then images were captured every 2.5 min in $35 \times 0.75\,\mu m$ z-sections for 2 h. All movies were deconvolved using the LIGHTNING adaptive approach and collapsed into 2D using a maximum intensity projection. BUBR1 foci number and intensity were determined using FIJI.

For live-cell imaging of PA-GFP-α-Tubulin, cells were seeded into 4-chamber, 35-mm glass-bottom culture dishes (Greiner) and maintained at 37 °C and 5% $CO_2$ in an environmental control station. Imaging was performed using a Lecia SP-8 confocal microscope, equipped with a resonance scanner, and 405 nm, 488 nm, 552 nm, and 638 nm laser lines. Images were acquired

with a $63 \times 1.4$ NA oil objective. Metaphase cells were identified using H2B-iRFP signal or differential inference contrast. The region immediately distal to the metaphase plate was marked for photoactivation. One pre-photoactivation frame ($4 \times 1\,\mu m$ z-sections) was collected, immediately followed by PA-GFP-α-Tubulin photoactivation using 5 pulses of the 405 nm laser at 50% intensity. Frames were then collected as before every 10 s for 6 min. Cells were excluded from analysis if they entered anaphase during the time-lapse. Spindle pole position was manually annotated in maximum intensity projected 2D time-lapse images, and measurements for GFP intensity were taken along the spindle pole axis in FIJI. Downstream analysis of tubulin dynamics was managed in R. Before all analyses, the non-photoactivated side of the spindle was used as background subtraction for the activated side. For flux measurements, the location of the maximum GFP intensity band along the spindle pole axis was determined in an automated fashion from smoothed intensity data.

## Cell growth and viability techniques

To measure 5-day endpoint cell growth and viability, cells were plated in triplicate for each condition in 200 μL media in a 96-well plate (RPE1 at 600 cells/well; DLD1, HCT116, HCC1806, HeLa, MCF10A, and OVCAR-8 at 1200 cells/well; MCF7 at 5000 cells/well; OVCAR-3 at 8500 cells/well; and MDA-MB-157 at 15,000 cells/well). The following day, drugs were added as indicated and cells were cultured for 5 days. To assay viability, 10 μL of 5 mg/mL Thiazolyl blue tetrazolium bromide (MTT; Abcam) solution in PBS was added to each well and incubated for 4.5 h. In total, 100 μL 10% SDS, 0.01 M HCl solution was then added to each well to dissolve the formazan crystals. Absorbance was measured with a spectrophotometer at 570 nm. % Viability = $(\text{Signal}_{[\text{sample}]} - \text{Signal}_{[\text{DMN}]})/(\text{Signal}_{[\text{DMSO}]} - \text{Signal}_{[\text{DMN}]}) \times 100$ unless specifically noted. % Toxicity = 100% − Viability. % Viability rescue = $(\text{Signal}_{[\text{sample, KIF18Ai}]} - \text{Signal}_{[\text{WT, KIF18Ai}]})/(\text{Signal}_{[\text{WT, DMSO}]} - \text{Signal}_{[\text{WT, KIF18Ai}]}) \times 100$. Drug titration curves were derived using the equation for "[Inhibitor] vs. response – Variable slope (four-parameter)" in GraphPad Prism 7 for Mac OS X (GraphPad Software, La Jolla, USA).

For longitudinal live-cell confluence and growth measurements, cells were plated in triplicate for each condition in 200 μL media in a 96-well plate (RPE1 at 600 cells/well; DLD1, HCT116, HCC1806, HeLa, MCF10A, and OVCAR-8 at 1200 cells/well; MCF7 at 5000 cells/well; OVCAR-3 at 8500 cells/well; and MDA-MB-157 at 15000 cells/well). The following day, drugs were added, and confluency measurements were taken using the CellCyte X live-cell incubator imaging system (Echo) every 4 h for 7 days.

## Flow cytometry

DNA ploidy analysis was assessed using propidium iodide (PI) staining. Cells were trypsinized and washed with 1% BSA in PBS (1500 rpm, 5 min) before being fixed in 70% ethanol. Following three washes with 1% BSA in PBS, a 20 min incubation at 37 °C in a solution of 10 μg/mL PI and 0.1 mg/mL RNaseA in PBS was performed. Samples were analyzed using a FACSCalibur Flow Cytometer (BD Biosciences) and data processing was done using FlowJo software.

## scDNA sequencing

Single-cell DNA sequencing was performed on RPE1, HCT116, and MCF10A cells grown for 7 days in DMSO or KIF18Ai for 7 days. Sequencing was performed on whole cells including micronuclei as described previously(van den Bos et al, 2019), and analyzed with the AneuFinder algorithm (Bakker et al, 2016).

## CRISPR–Cas9 genome-wide screens

Pooled genome-wide CRISPR–Cas9 knockout screens in HCC1806 and OVCAR-3 cells were performed as described previously (Chen et al, 2015; Lambrus et al, 2016; Shalem et al, 2014). HCC1806 cells were infected with lentiCas9-Blast (Addgene; #52962). OVCAR-3 were infected with a lentiviral construct containing Cas9 and a Hygromycin resistance cassette. Positive selection of transduced cells was performed 2 days post-transfection with 400 μg/mL Hygromycin or 5 μg/mL blasticidin, respectively. Monoclonal cell lines were isolated by limiting dilution and Cas9 expression was validated by immunoblotting.

The human Brunello CRISPR knockout sgRNA library was purchased from Addgene (a gift of David Root and John Doench; #73178) and plasmid DNA amplified according to the manufacturer's instructions. To produce virus, the Brunello pooled plasmid library and the lentiviral packaging plasmids psPAX2 and pMD2.G were co-transfected into $40 \times 15$-cm culture dishes of HEK293FT cells. $6 \times 10^6$ HEK293FT cells were seeded into a poly-l-Lysine-coated 15 cm culture dish the day before transfection. For each 15 cm dish, the following DNA was diluted in 1.2 mL OptiMEM (Thermo Fisher Scientific): 9 μg lentiviral vector, 12 μg psPAX2, and 3 μg pMD2.G. Separately, 70 μl of 1 μg/μL 25-kD polyethylenimine (PEI) (Sigma-Aldrich) was diluted into 1.2 mL OptiMEM and incubated at room temperature for 5 min. After incubation, the DNA and PEI mixtures were combined and incubated at room temperature for 20 min. During this incubation, the culture media was replaced with 16 mL pre-warmed DMEM + 1% FBS. The transfection mixture was then added dropwise to the 15 cm dish. Viral particles were harvested at 24, 48, and 72 h after the media change. Media collected from 24, 48, and 72 h were pooled and filtered through a 0.45-μm PVDF syringe filter. The media was then concentrated using Amicon Ultra-15 Centrifugal Filter Unit with Ultracel-50 membrane (EMD Millipore Corporation cat# UFC905024). The virus was then frozen and stored at −80 °C.

Cells were transduced with the Brunello library via spinfection as previously described (Lambrus et al, 2016). To determine the optimal virus volumes for achieving an MOI ~0.3, each new batch of virus was titered by spinfecting $3 \times 10^6$ cells with several different volumes of virus. Briefly, $3 \times 10^6$ cells per well were seeded into a 12-well plate in growth media supplemented with 10 μg/mL polybrene. Each well-received a different titrated virus amount (between 5 and 50 μL) along with a no-transduction control. The plate was centrifuged at $1000 \times g$ for 2 h at 35 °C. After the spin, media was aspirated, and fresh growth media was added. The following day, cells were counted, and each well was split into duplicate wells. One well-received 3 μg/mL puromycin (Sigma) for 3 days. Cells were counted and the percent transduction was calculated as the cell count from the replicate with puromycin divided by the cell count from the replicate without puromycin

multiplied by 100. The virus volume yielding a MOI closest to 0.3 was chosen for large-scale transductions.

For the pooled screens, a theoretical library coverage of ≥250 cells per sgRNA was maintained at every step. Cells were infected at MOI ~0.3 and selected with puromycin at 3 μg/ml for 3 days. MOI was calculated using a control well infected in parallel following the procedure outlined above. Infected cells were expanded under puromycin selection for 6 or 7 days and subsequently seeded into 15-cm dishes a day prior to treatment. KIF18Ai doses that corresponded to individual LD80 for HCC1806 cells (25 nM) or OVCAR-3 (20 nM) cells were used while DMSO vehicle served as the negative control. Cell pallets were taken after 21 days for HCC1806 cells, 40 days for OVCAR-3 cells, and 4 days for OVCAR-8 cells, at which point the screen was terminated.

Genomic DNA was isolated using the QIAamp Blood Maxi Kit (Qiagen) per the manufacturer's instructions. Genome-integrated sgRNA sequences for each sample were amplified and prepared for Illumina sequencing using a two-step PCR procedure as previously described (Lambrus et al, 2016). For the first PCR, a region containing the sgRNA cassette was amplified using primers specific to the sgRNA-expression vector (lentiGuide-PCR1-F: 5′-aatggac-tatcatatgcttaccgtaacttgaaagtatttcg-3′; lentiGuide-PCR1-R: 5′-ctttagtttgtatgtctgttgctattatgtctactattctttcc-3′). The thermocycling parameters for the first PCR were as follows: 98 °C for 1 min, 20 cycles of (98 °C for 30 s, 65 °C for 30 s, 72 °C for 30 s), and 72 °C for 1 min. The resulting amplicons for each sample were pooled and purified using AMPURE XP beads (Beckman Coulter) with a bead-to-sample ratio of 0.6x and 1.0x for double-size selection to exclude primers and genomic DNA. Primers for the second PCR include Illumina adapter sequences, a variable length sequence to increase library complexity, and a 8 bp barcodes for multiplexing of different biological samples (F2: 5′-aatgatacggcgaccaccgagatcta-cactctttccctacacgacgctcttccgatct-[4–7 bp random nucleotides]-[8 bp barcode]-tcttgtggaaaggacgaaacaccg-3′; R2: 5′-caagcagaagacggcatac-gagatgtgactggagttcagacgtgtgctcttccgatcttctactattctttcccctgcactgt-3′). In total, 2.5 μL of the product from the first PCR reaction was used, and the thermocycling parameters for the second PCR were as follows: 98 °C for 30 s, 12 cycles of (98 °C for 1 s, 70 °C for 5 s, 72 °C for 35 s). Second PCR products were pooled, and purified using AMPURE XP beads with a bead-to-sample ratio of 1.8× and quantified using the Qubit dsDNA BR Assay Kit (Thermo Fischer Scientific). Diluted libraries with 5% PhiX were sequenced with MiSeq (Illumina).

Sequencing data were processed for sgRNA representation using custom scripts. Briefly, sequencing reads were first demultiplexed using the barcodes in the forward primer and then trimmed to leave only the 20 bp sgRNA sequences. The spacer sequences were then mapped to the spacers of the designed sgRNA library using Bowtie (Langmead et al, 2009). For mapping, a maximum of one mismatch was allowed in the 20 bp sgRNA sequence. Mapped sgRNA sequences were then quantified by counting the total number of reads. The total numbers of reads for all sgRNAs in each sample were normalized.

We used the MaGeCK scoring algorithm (model-based analysis of genome-wide CRISPR–Cas9 knockout) to analyze and rank the genes from the screens (Li et al, 2014). β-scores and FDR values were also derived for each gene between DMSO or KIF18Ai conditions. Genes with a β-score outside 1.5 standard deviations from the population mean and with an FDR cutoff of ≤0.3 were

taken forward for further validation. Graphing and downstream analysis was performed in R. Gene ontology analysis was performed using the topGO algorithm (Alexa and Rahnenfuhrer, 2023).

CRISPR–Cas9 pooled, knockout screens in OVCAR-8 cells were performed by Cellecta, Inc. In brief, $1 \times 10^8$ OVCAR-8 Cas9 cells were infected with a proprietary lentiviral sgRNA library at MOI 0.4–0.5 and cultured for 4 days. Transduced cells were then selected with puromycin for 3 days. Two days later, samples were treated with either 11 nM KIF18Ai ($IC_{50}$), 33 nM KIF18Ai ($IC_{90}$) or DMSO. Cells were then cultured for 4 days before DNA isolation. sgRNA abundance in each population was determined using deep sequencing. β-scores and FDR values between DMSO and KIF18Ai conditions were used for downstream analysis. Genes with a β-score outside 1.5 standard deviations from the population mean and with an FDR cutoff of ≤0.1 were taken forward for further validation. Graphing and downstream analysis was performed in R. Gene ontology analysis was performed using the topGO algorithm (Alexa and Rahnenfuhrer, 2023).

## Data availability

The single-cell whole-genome sequencing data from this publication have been deposited to the ENA database (https://www.ebi.ac.uk/ena) and assigned the identifier PRJEB69520.

## Peer review information

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

## Acknowledgements

This work was supported by the National Institutes of Health grants R01GM114119, R01GM133897 and R01CA266199 (to AJH), and a research agreement sponsored by Amgen Inc. (to AJH). We also would like to acknowledge the use of the Johns Hopkins Flow Cytometry Shared Resource Center.

## Author contributions

**Colin R Gliech**: Conceptualization; Data curation; Software; Formal analysis; Validation; Investigation; Visualization; Methodology; Writing—original draft; Writing—review and editing. **Zhong Y Yeow**: Data curation; Formal analysis; Investigation; Methodology. **Daniel Tapias-Gomez**: Data curation; Formal analysis; Investigation; Methodology. **Yuchen Yang**: Data curation; Formal analysis; Investigation; Methodology. **Zhaoyu Huang**: Data curation; Formal analysis; Investigation; Methodology. **Andréa E Tijhuis**: Data curation; Formal analysis; Investigation; Visualization; Methodology. **Diana CJ Spierings**: Data curation; Formal analysis; Investigation; Methodology. **Floris Foijer**: Conceptualization; Data curation; Formal analysis; Supervision; Visualization; Methodology. **Grace Chung**: Conceptualization; Data curation; Formal analysis; Investigation; Methodology. **Nuria Tamayo**: Conceptualization; Data curation; Formal analysis; Investigation; Methodology. **Zahra Bahrami-Nejad**: Conceptualization; Data curation; Formal analysis; Investigation; Methodology. **Patrick Collins**: Conceptualization; Data curation; Formal analysis; Investigation; Methodology. **Thong T Nguyen**: Conceptualization; Data curation; Formal analysis; Investigation; Methodology. **Andres Plata Stapper**: Conceptualization; Data curation; Formal analysis; Investigation; Methodology. **Paul E Hughes**: Conceptualization; Resources; Supervision; Project administration. **Marc Payton**: Conceptualization; Resources; Supervision; Project administration. **Andrew J Holland**: Conceptualization; Resources; Software; Formal analysis; Supervision; Funding acquisition; Methodology; Writing—original draft; Project administration; Writing—review and editing.

## Disclosure and competing interests statement

The authors declare no competing interests.

# Expanded View Figures

**Figure EV1.  Extended analysis of cellular response to KIF18Ai.**

(**A**) Wide-field immunofluorescence of KIF18A localization relative to spindle poles (CEP192, yellow) and the mitotic spindle (α-Tubulin, magenta) in response to KIF18Ai treatment across full cell line panel. Scale bar = 5 µm. (**B**) Titration of KIF18Ai in a 5-day MTT endpoint viability assay for the panel of sensitive and insensitive cell lines. $N \geq 3$ independent experiments, $n \geq 2$ technical replicates per experiment. Data are represented as mean ± SEM. (**C**) Western blot of KIF18A expression levels across the full panel of sensitive and insensitive cell lines. (**D**) Linear correlation between 5-day KIF18Ai toxicity and normalized KIF18A expression (KIF18A/α-Tubulin) from (**C**). Expression is plotted on a $\log_2$ axis. (**E**) Linear correlation between 5-day KIF18Ai toxicity and Nutlin-3 toxicity from a 5-day MTT endpoint viability assay for the panel of sensitive and insensitive cell lines. $N = 3$ independent experiments, $n = 3$ technical replicates per experiment. Data are represented as mean ± SD.

▶

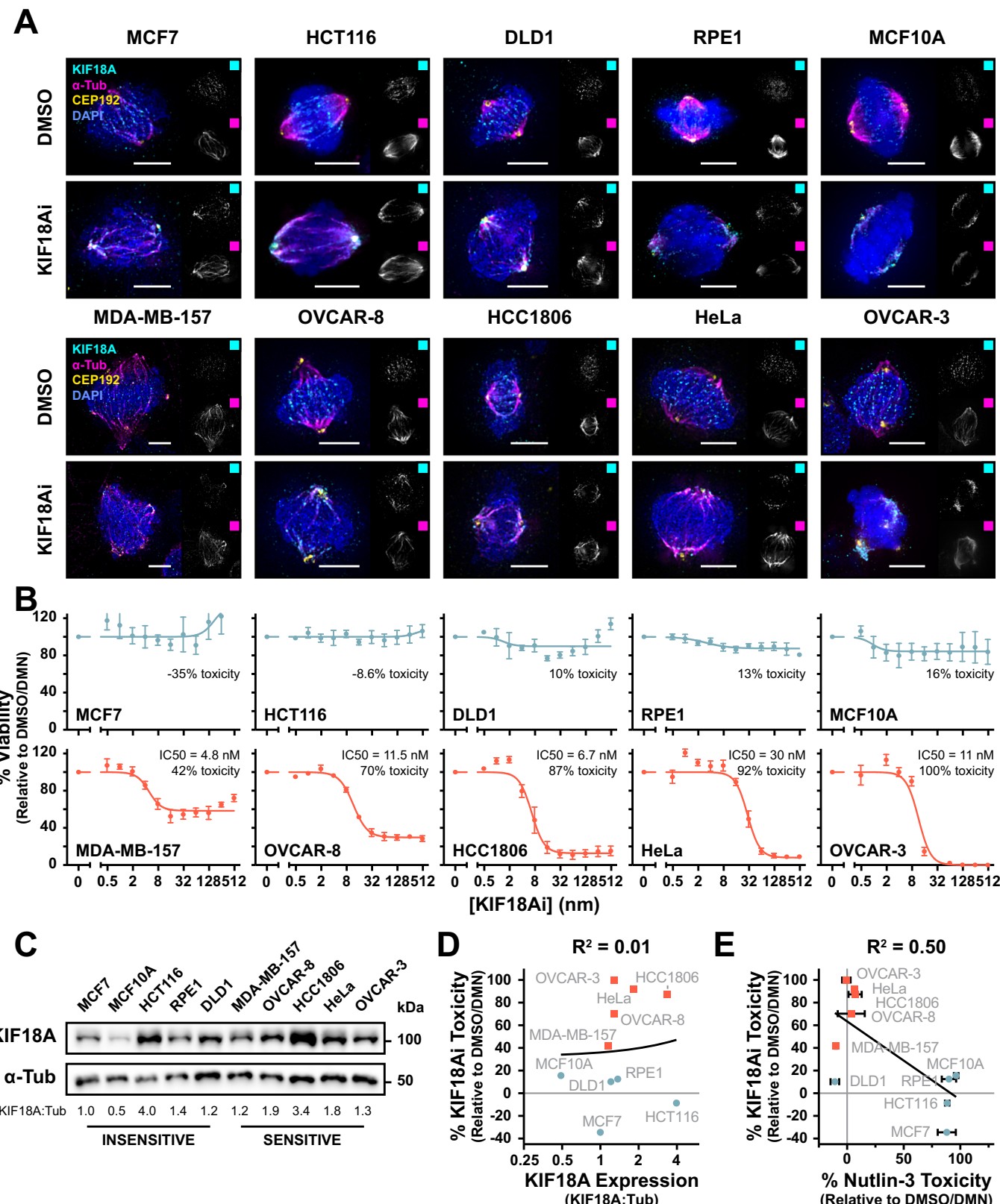

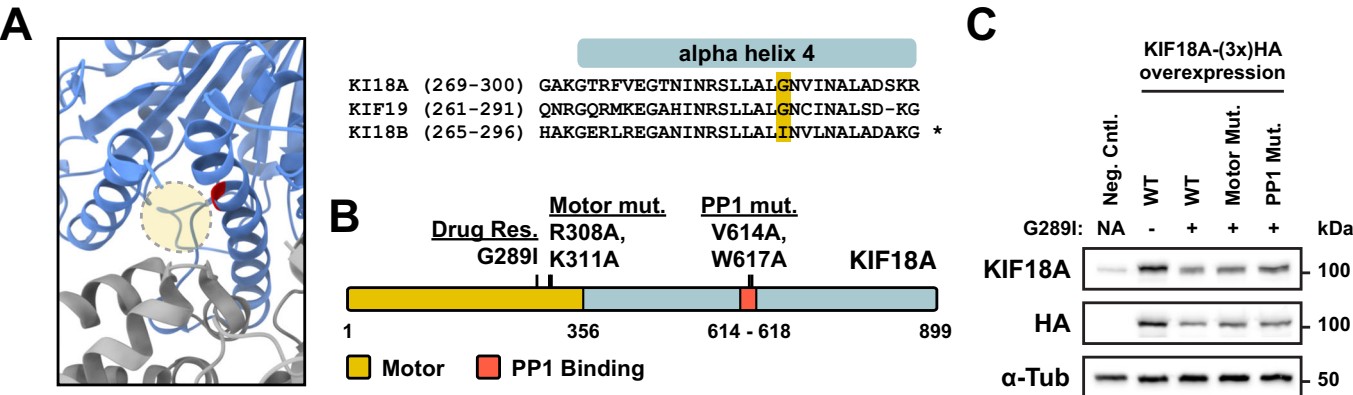

**Figure EV2. Validation of drug-resistant KIF18A transgenic HCC1806 cells.**

(A) Schematic of KIF18Ai drug binding site from EM structure (PDB: 5OAM). Blue: KIF18Ai motor domain with G289 highlighted in red. Gray: α-tubulin/β-tubulin. Highlighted circle represents drug binding pocket. Protein sequence alignment of drug binding pocket between KIF18A, KIF19, and KIF18B. (B) Schematic of KIF18A protein domains and mutations. (C) Western blot validation of HCC1806 cells constitutively expressing a WT, drug-resistant (G289I), drug-resistant and motor dead (G289I, R308A, K311A), or drug-resistant and PP1-binding deficient (G289I, V614A, W617A) KIF18A-(3x)HA transgene.

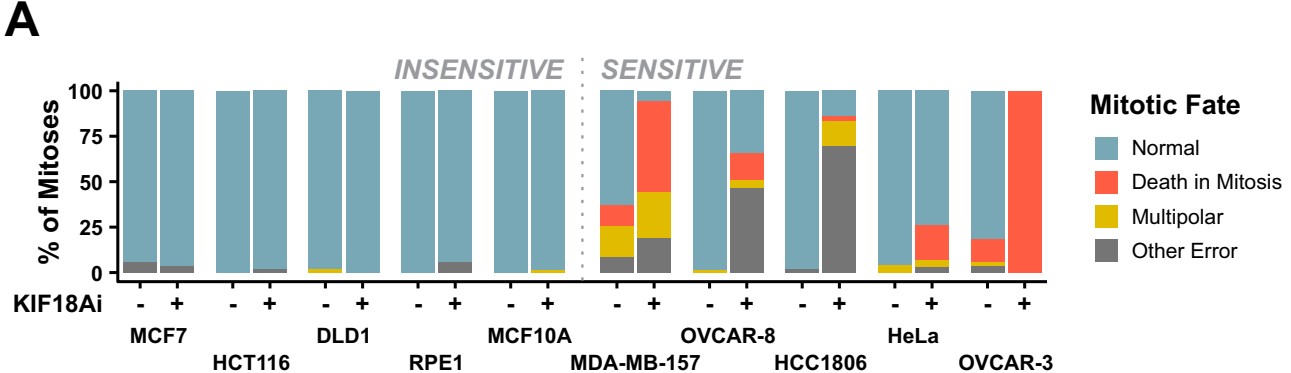

**Figure EV3.   Live-cell analysis of mitotic outcomes from KIF18Ai Treatment.**

(A) Proportion of mitotic fates across the panel of H2B/α-Tubulin fluorescently tagged cell lines in response to DMSO or KIF18Ai treatment from live-cell wide-field time-lapse microscopy in Fig. 1E. (B) Representation of mitotic error threshold across a panel of H2B/α-Tubulin fluorescently tagged cell lines in response to DMSO or KIF18Ai treatment from live-cell wide-field time-lapse microscopy in Fig. 1E. Bars represent individual mitotic events. Dotted line represents error threshold.

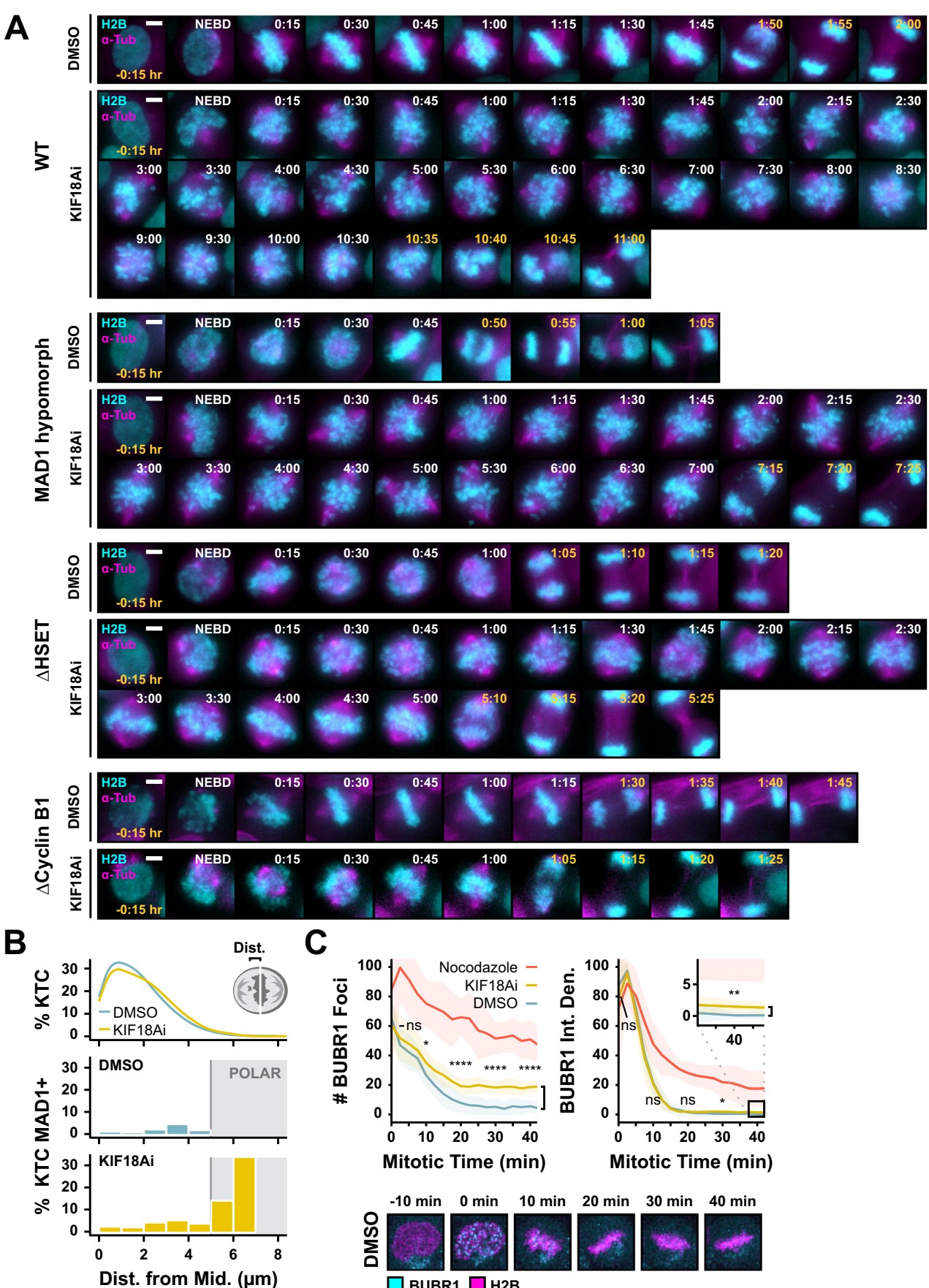

◀   **Figure EV4.  Extended chromosome alignment and SAC activation data in KIF18Ai.**

(A) Representative live-cell time-lapse microscopy stills from HCC1806 cells used to evaluate mitotic chromosome congression. Scale bar = 5 µm. (B) Top: Kinetochore distribution probability along the spindle axis from Fig. 3C. Bottom: Probability of a kinetochore being MAD1+ relative to its position on the spindle axis in DMSO or KIF18Ai conditions. Polar kinetochores are defined as >5 µm from the spindle midline. (C) Top: Quantification of BUBR1 foci from live-cell confocal time-lapse microscopy of HeLa EGFP-BUBR1 BAC H2B-iRFP cell lines in DMSO, KIF18Ai, and Nocodazole conditions. Error bars represent mean ± SD. Statistical significance was determined using an unpaired two-tailed Student's *t* test between DMSO and KIF18Ai conditions at 0, 10, 20, 30, and 40 min. Sample size and full statistical results are listed in Dataset EV2. Bottom: Representative still images from mitotic movies. *$P < 0.05$, **$P < 0.01$, ***$P < 0.001$, and ****$P < 0.0001$.

                                          

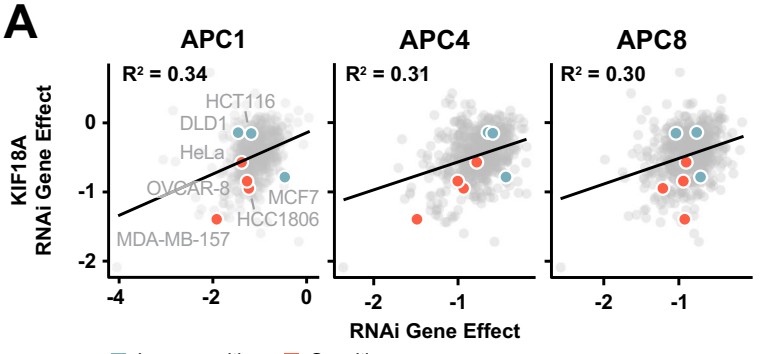

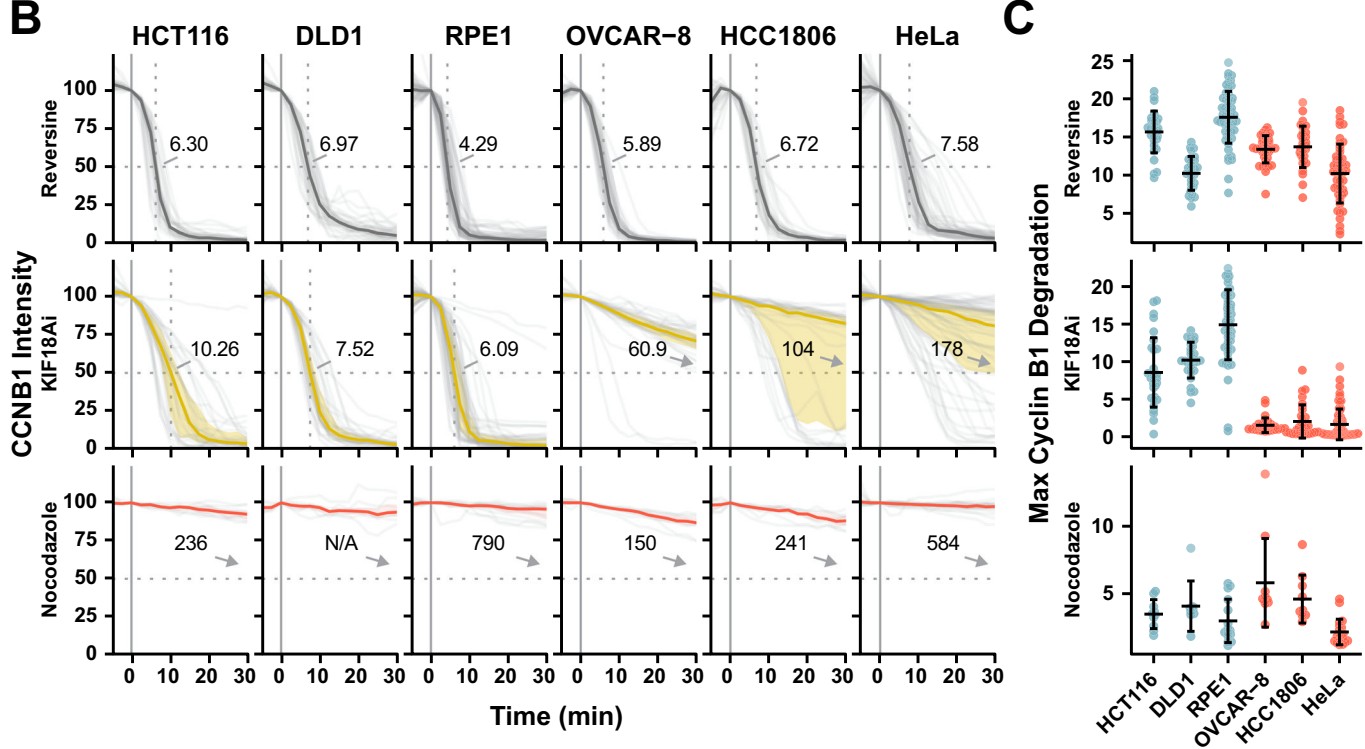

**Figure EV5. Extended analysis of APC/C activity relative to KIF18A dependency.**

(A) Top 10 RNAi co-dependency relationships from DepMap dataset for KIF18A. Red and blue points represent sensitive and insensitive cell lines respectively from the panel in Fig. 1C. Bolded table entries are APC/C or SAC genes. (B) Quantification of Cyclin B1 degradation rates for endogenously tagged fluorescent cells treated with Reversine, KIF18Ai, or Nocodazole. The median (colored line) of individual traces (gray lines) is plotted, with the shaded region encompassing the first and third quartile of each population. Cyclin B1 signal is normalized to the metaphase inflection point and median $t_{1/2}$ values are listed. Full sample size information is listed in Dataset EV3. (C) Maximum slope of Cyclin B1 degradation at metaphase for endogenously tagged fluorescent cells treated with Reversine, KIF18Ai, or Nocodazole in (B). Rates are calculated relative to the total Cyclin B1 signal at mitotic entry. Data are represented as mean ± SD. $N \geq 20$ cells per condition (KIF18Ai, Reversine), $N \geq 5$ cells per condition (nocodazole).

