## [Peer Review File · The EMBO Journal]

Weakened APC/C activity at mitotic exit drives cancer vulnerability to KIF18A inhibition

Colin Gliech, Zhong Yeow, Daniel Tapias-Gomez, Yuchen Yang, Zhaoyu Huang, Andrea Tjihuis, Diana Spierings, Floris Fojjer, Grace Chung, Nuria Tamayo, Zahra Bahrami-Nejad, Patrick Collins, Thong Nguyen, Andres Plata Stapper, Paul Hughes, Marc Payton, and Andrew Holland

DOI: [10.15252/emj.2023114411](https://doi.org/10.15252/emj.2023114411)

Corresponding author(s): Andrew Holland (aholland@jhmi.edu)

Review Timeline:

Submission Date:	1st May 23
Editorial Decision:	12th Jun 23
Revision Received:	10th Nov 23
Editorial Decision:	14th Dec 23
Revision Received:	23rd Dec 23
Accepted:	2nd Jan 24

Editor: Hartmut Vodermaier

Transaction Report:

Dr. Andrew J Holland
Johns Hopkins School of Medicine
Dept of Molecular Biology & Genetics
725 N Wolfe St, 704 PCTB
Baltimore, orcid||||| 21205

12th Jun 2023

Re: EMBOJ-2023-114411
SAC activation in cells with reduced APC/C activity drives cancer vulnerability to KIF18A inhibition

Dear Andrew,

Thank you again for submitting your manuscript for consideration by The EMBO Journal, and apologies for the delay in getting back to you with a response - the final reports came in only while I was out of office on vacation. Please find the three sets of referee comments now copied below in this message.

As you will see, all referees consider the topic of the investigation of general interest both from the biological and the therapeutic aspect, but remain a bit divided in their views on the significance and unexpectedness of the key conclusions of the work. Moreover, reviewers 2 and 3 raise a number of specific issues, addressing which would be important in order to strengthen the conclusiveness of the study.

Given the overall interest of the subject combined with the unbiased and comprehensive approach, we decided that we would remain open to publishing a revised version of this study; however, since it is our policy to consider only a single round of major revision, it would clearly be helpful to clarify how each of the criticisms/queries (esp. those raised by reviewers 2 and 3) might be answered. I would therefore invite you to consider the referees' comments together with your coworkers, and to send me a tentative response letter, which we could then further discuss via email or a follow-up video call, in order to decide on the most viable options here. I should add that we could also offer extension of the default three-months revision period if needed, with our 'scooping protection' (meaning that competing work appearing elsewhere in the meantime will not affect our considerations of your study) remaining of course valid also throughout this extension.

Detailed information on preparing, formatting and uploading a revised manuscript can be found below and in our Guide to Authors. Thank you again for the opportunity to consider this work for The EMBO Journal, and I look forward to hearing from you in due time.

With kind regards,

Hartmut

- 3) Revised manuscript text (including main tables, and figure legends for main and EV figures) has to be submitted as editable text file (e.g., .docx format). We encourage highlighting of changes (e.g., via text color) for the referees' reference.
- 4) Each main and each Expanded View (EV) figure should be uploaded as individual production-quality files (preferably in .eps, .tif, .jpg formats). For suggestions on figure preparation/layout, please refer to our Figure Preparation Guidelines: <http://bit.ly/EMBOPressFigurePreparationGuideline>
- 5) Point-by-point response letters should include the original referee comments in full together with your detailed responses to them (and to specific editor requests if applicable), and also be uploaded as editable (e.g., .docx) text files.
- 6) Please complete our Author Checklist, and make sure that information entered into the checklist is also reflected in the manuscript; the checklist will be available to readers as part of the Review Process File. A download link is found at the top of our Guide to Authors: embopress.org/page/journal/14602075/authorguide
- 7) All authors listed as (co-)corresponding need to deposit, in their respective author profiles in our submission system, a unique ORCID identifier linked to their name. Please see our Guide to Authors for detailed instructions.
- 8) Please note that supplementary information at EMBO Press has been superseded by the 'Expanded View' for inclusion of additional figures, tables, movies or datasets; with up to five EV Figures being typeset and directly accessible in the HTML version of the article. For details and guidance, please refer to: embopress.org/page/journal/14602075/authorguide#expandedview
- 9) Digital image enhancement is acceptable practice, as long as it accurately represents the original data and conforms to community standards. If a figure has been subjected to significant electronic manipulation, this must be clearly noted in the figure legend and/or the 'Materials and Methods' section. The editors reserve the right to request original versions of figures and the original images that were used to assemble the figure. Finally, we generally encourage uploading of numerical as well as gel/blot image source data; for details see: embopress.org/page/journal/14602075/authorguide#sourcedata

At EMBO Press, we ask authors to provide source data for the main manuscript figures. Our source data coordinator will contact you to discuss which figure panels we would need source data for and will also provide you with helpful tips on how to upload and organize the files.

In the interest of ensuring the conceptual advance provided by the work, we recommend submitting a revision within 3 months (10th Sep 2023). Please discuss the revision progress ahead of this time with the editor if you require more time to complete the revisions. Use the link below to submit your revision:

Link Not Available

Referee #1:

General summary and opinion about the principal significance of the study, its questions and findings

In this manuscript, Glietch et al perform a detailed mechanistic analysis of a recently developed KIF18A inhibitor, which displays preferential toxicity in cancer lines. This work is important and timely because recent data demonstrated that cancer cells with tetraploidy or chromosomal instability (CIN) are more sensitive to KIF18A knockdown/knockout, implying that small molecule inhibitors targeting KIF18a may be useful for treating a wide range of cancer types. To facilitate effective use of these drugs in the clinic, it is crucial to understand the molecular basis for KIF18a sensitivity across cell types.

This manuscript starts by performing a thorough characterisation of a panel of cell lines (non-transformed and different cancer types) that are sensitive and insensitive to KIF18A inhibition. CRISPR screening is used to identify different resistance mechanisms, and these are independently validated for three different targets. These studies led to the conclusion that KIF18A inhibition affects chromosome oscillations and kinetochore attachments to produce a SAC dependent delay, which ultimately arrests cells in mitosis and leads to toxicity. A further CRISPR screen led to the discovery that reduced APC activity sensitizes cells to KIF18A inhibition by enhancing the SAC-dependent delay. Importantly, this appears to be a strong predictor of KIF18a sensitivity, because mitotic and metaphase duration correlated strongly with sensitivity across the panel of tested cell types, and

DepMap analysis implies this relationship holds true across many more cancer lines. Finally, analysis of live cyclin B degradation assays demonstrate sensitivity is directly associated with reduced APC activity.

In summary, the authors have performed very careful and detailed mechanistic analysis, which I found to be sufficient to fully support their conclusions. I feel that the most significant conclusion is that reduced APC/C activity, which may be a common feature of cancer cells, causes sensitivity to KIF18a inhibition by magnifying the SAC-dependent delays. Another important conclusion is that elevated chromosome numbers can also magnify these delays by directly increasing the total SAC signal, and that these two features together can synergise to enhance KIF18i-sensitivity. These conclusions are particularly significant because cancer cells with CIN frequently display elevated chromosome numbers, and partial APC/C inhibition has been implicated as a common mechanism that cancer cells use to tolerate CIN. Therefore, cancer cells could well have evolved the two main features that give rise to KIF18A sensitivity. This study therefore lays the foundations to develop biomarkers that can assess these features across different cancer lines, and ultimately, patient tumours.

Specific major concerns essential to be addressed to support the conclusions

I have no major concerns

Minor concerns that should be addressed

- 1) I cannot find the legends that describes the details/titles of Movies S1 - S5?
- 2) In relation to: "Finally, we find that the degree of SAC signaling does not correlate well with sensitivity"
- This should read "the degree of BUBR1 and MAD1 recruitment does not correlate well with sensitivity. SAC signaling encompasses much more than just the recruitment of these proteins to kinetochores (e.g. CDC20 recruitment and phosphorylation changes on MAD1, BUB1).
- 3) in relation to: "these drugs have failed to progress in the clinic."
- Some are still in clinical trials, so it is perhaps better to state that they have "so far failed to progress in the clinic."
- 4) In relation to: "However, a G289I KIF18A mutant unable to bind PP1 largely rescued growth (Figures 2H and S2B)."
- I think the referenced figure should be 2I.
- 5) A very recent manuscript from Tsang and Cheeseman provides a potential explanation for differences in mitotic cell fate following a prolonged mitotic arrest (relating to difference in CDC20 variants). This could perhaps be mentioned in the relevant section of the discussion.

Any additional non-essential suggestions for improving the study (which will be at the author's/editor's discretion)

It is interesting that KT-MT turnover is more strongly affected by KIF18 inhibition in sensitive HCC1806, in comparison to insensitive RPE1 cells (Fig.3I). Is this a common feature that could also explain KIF18a sensitivity across more cell lines? It may be challenging to perform detail photo-activation experiments in multiple cell types, but the effects of rapid destabilisation of KT-MT with cold-shock +/- KIF18A inhibition may reveal important differences in sensitive and insensitive cell types.

In relation to the above, I appreciate that SAC strength/duration of mitosis is an important predictor of outcome, but should the direct sensitivity to KT-MT destabilisation following KIF18i not also be highlighted as a potential mechanism of sensitivity? It certainly tracks with HCC1806 sensitivity (-/+ HSET knockout), so is there a reason to believe this will not also be the case in other sensitive lines? I understand that cell lines that are sensitive for this reason would be predicted to also display higher BUBR1/MAD1 signals, but could this explain sensitivity in MDA-MB-157 cells, for example, which do have high BUBR1/MAD1?

Adrian T Saurin

Referee #2:

Accurate chromosome segregation requires the successful attachment of chromosomes via their kinetochores to the mitotic spindle. This process is modulated by a range of enzyme and non-enzyme proteins localized both to spindle microtubules and kinetochores. Recently, the mitotic motor protein KIF18A, necessary for normal chromosome alignment during mitosis, has attracted a lot of attention because it was found to be specifically required for successful cell division in cells with abnormal

ploidies, suggesting that it may constitute a useful target in the therapy of tumours characterised by aneuploidy. However, despite these important findings, it has so far remained elusive which molecular differences determine the differential cell viability upon loss of KIF18A.

In this manuscript Glietch and colleagues analyse the response of a panel of non-transformed and transformed cell lines to a recently obtained small molecule inhibitor of KIF18A (referred to as KIF18Ai). Informed by cell viability data in the presence or absence of the drug, the authors split their panel of cells into KIF18Ai-sensitive and insensitive cell lines and then try to identify key factors for the differential outcome. Using a combination of CRISPR/Cas9 knockout screens in KIF18Ai-sensitive and partially sensitive cell lines and cell biological analysis of microtubule-kinetochore attachment formation, spindle assembly checkpoint function and activity of the anaphase promoting complex/cyclosome (APC/C) the authors come to the conclusion that the determining factor for KIF18Ai-sensitivity is the precise balance of spindle checkpoint signalling versus anaphase promoting complex/cyclosome (APC/C) activity in different cell lines.

The question of why loss or inhibition of KIF18A is toxic in some cell lines but not in others is very interesting. While this manuscript attempts to uncover the molecular biology governing the response to loss of KIF18A, the results are not particularly enlightening. This is partially due to the dense and confusing lay-out of the paper which presents a large number of experiments looking at different aspects of chromosome segregation process in genetically different cell lines without a clear rationale of why some experiments are conducted in one cell line while others are carried out in a different cell line, and without providing real depth to any single point of investigation. Most of the data are buried in difficult to interpret graphs, preventing the reader to assess the quality of the raw data. The constant change between cell lines is extremely confusing. The manuscript contains many quite general statements (e.g. "We conclude that hyperploidy and low APC/C activity act synergistically to generate KIF18A dependency.") when the data were only obtained with only one of the cell lines (in this example RPE1 cells). Given that the authors set out to identify more general determinants of KIF18Ai sensitivity, their experiments should be repeated with the entire panel of cell lines before such statements are made. There is also a degree of uncertainty as to what the authors are actually investigating. In their first experiments they look at cell viability 5 days after exposure of cells to KIF18Ai. There have recently been several publications (some from the authors' own lab) demonstrating that perturbed mitosis can lead to p53-dependent cell cycle arrest or cell death in the following G1 phase. The p53 status of the cell lines the authors are investigating is therefore very important for the interpretation of the results and should be disclosed in the form of a table in Figure 1.

In the rest of the manuscript, the authors focus mainly on the direct consequences of KIF18Ai for mitotic progression without mentioning or discussing that the loss of cell viability may be an indirect effect of the cell-line specific response to mitotic perturbations in the next G1 phase, not the mitotic perturbation in itself. This is particularly surprising because their own data show that the mitotic response of the cell lines does not seem to be the differentiating factor ("Unexpectedly, we also found that both sensitive and insensitive cell lines experienced similar increases in the per-kinetochore BUBR1 signal and the number of MAD1+ kinetochores per cell in response to KIF18Ai (Figures 4A-D). Ultimately, this reveals that SAC signaling at kinetochores is a poor overall predictor of KIF18Ai sensitivity (Figures 4B and 4D)."). In other words, are p53 (functionally) negative HeLa cells sensitive to KIF18Ai because they undergo defective mitosis and don't stop cell proliferation in response to this and then die, whereas p53 wild type RPE1 cells enter a prolonged G1 arrest in response to minor mitotic disturbances caused by KIF18Ai and hence survive? There are some live cell imaging experiments of cells exposed to KIF18Ai (e.g. Figure 1E), which may be interesting in this regard, but from the way the data are presented it is difficult to find out how many of the cell stop dividing during the course of the experiments. However, from the numbers in Supplemental table 1 it seems that far fewer mitotic events were recorded in RPE1 cells after exposure to KIF18Ai than without, in line with the idea that these cells stopped proliferating. In summary, in the current form, the manuscript does not clearly differentiate between direct mitotic defects incurred by KIF18Ai and the cellular response to defective mitosis, and despite the authors' goal of defining the molecular biology that render cells sensitive to KIF18Ai generally, the results that the authors report appear to be specific to certain cell lines.

Specific comments:

- Figure 1: The reason for the choice of cell lines should be more clearly articulated, and a table should be provided to specify p53 status, ploidy and origin of the chosen cell lines. HeLa is not an ovarian cancer!! Since ploidy has previously been described as a differentiating factor for sensitivity or insensitivity to KIF18A depletion it is particularly important that the cell lines are classified according to this. How do the authors findings tie in with the idea that ploidy is the differentiating factor for sensitivity to loss of KIF18A?
- It is extremely confusing that not all of the experiments are conducted in all cell lines, and there is no proper explanation of why some experiments are conducted in one cell line and not another. Given that the authors argue that their results may be important clinically, it is critical that their conclusions are checked in all of their cell lines, and that the genetic background (transformed/untransformed, p53 status, ploidy etc.) is taken into consideration and discussed when the results are interpreted.
- Figure 1A: The authors state "Treatment with KIF18Ai causes stable binding of the KIF18A motor domain to microtubules (Tamayo et al., 2022), leading to the relocation of KIF18A in mitosis from kinetochores to spindle poles in all the cell lines tested (Figures 1A and S1A). This mis-localization phenocopies the effect of KIF18A loss (Payton et al., 2023), likely by blocking KIF18A-induced stabilization of microtubule plus ends (Du et al., 2010)." This statement does not make any sense - surely in cells without KIF18A, there cannot be mis-localisation of KIF18A? Also, the referenced paper "Payton et al, 2023" is under review according to the reference list (and does not appear to be on bioRxiv) so cannot be cited as evidence, as the reader has no access to this information.
- Figure 3: "Knockout of HSET and, to a lesser extent Cyclin B1, dramatically improved the ability to form and maintain a metaphase plate in KIF18Ai (Figure 3B)." This information is not visible in Figure 3B. Could the authors please show representative time lapse images instead of the very confusing graph? The information that is easily extracted from the graph is that mitotic duration is drastically reduced in cells lacking cyclin B1 but that is much more likely due to an abrogation of mitosis

altogether rather than an improvement of the cells' ability to hold a metaphase plate. Notably, this graph lacks a WT control.

- Figure 3: The cell biological analysis presented in the manuscript relating to the stability of microtubule-kinetochore attachments needs to be improved. The only experiment that was conducted to demonstrate a defect in microtubule-kinetochore attachment stability upon KIF18Ai is the decay of photo-activatable GFP-alpha-tubulin (Figure 4). This was only done in one "insensitive" cell line (RPE1) and one "sensitive" cell line (HCC1806) so does not give any information about whether any differences are reproducible for the other insensitive or sensitive cell lines. Furthermore, this assay is difficult to interpret because of the inherently different rates of microtubule flux in the different cell lines. A much more informative assay would be the classical cold-stability assay combined with the measurement of inter-kinetochore stretch. As these assays do not rely on genetic modifications of the cell line to be investigated, they can easily be done on a panel of different cell lines and would hence lead to comparable results. As it stands, the statement that "Taken together, our data show that treatment with KIF18Ai leads to a weakening of K-MT attachments...." is not sufficiently supported by data.

- Figure 6: In this figure, the authors conclude from their cyclin B degradation assays that HeLa cells have inherently lower APC/C activity than RPE1 cells and that this is the reason that HeLa cells are sensitive to KIF18Ai. The authors then go on to test this hypothesis by overexpressing the ubiquitin E2 ligase UBE2S to increase APC/C activity, but surprisingly and nonsensically, do not carry out this experiment in HeLa cells but in HCC1806 cells. Why was the experiment not performed in HeLa cells (or indeed all cell lines of their panel), and could this experiment please be conducted?

Referee #3:

This manuscript from Gliech and colleagues investigates the molecular determinants of cancer cell dependency on the kinesin KIF18A. Previous screens identified KIF18A as being uniquely essential in a subset of cancer cells. This prior work generated interest in KIF18A as a target for cancer therapy, and several pharmaceutical companies (including Amgen- which employs authors on this paper) have announced programs focused on developing and testing anti-KIF18A inhibitors. However, what the underlying differences are that confer sensitivity to KIF18A in some cancer cells remains an important, unanswered question.

To address this, the authors performed whole genome CRISPR screens to identify genes that enhance or suppress the effects of a KIF18A inhibitor on proliferation of HCC1806 and OVCAR-8 cells. The top hits from the screens have roles in the SAC and regulation of microtubules. The authors provide detailed characterizations of cellular phenotypes following combined KIF18A inhibition and knockout/ knockdown of MAD1, HSET, or Cyclin-B. The data presented are impressive and thorough. However, the conclusions are not particularly surprising in the context of prior work though. For example, previous studies have demonstrated that the mitotic arrest and loss of viability seen after inhibition of KIF18A function depend on the SAC. Given this information, and the known relationship between the SAC and the APC, it is also not surprising that inhibition of the APC enhances the arrest caused by KIF18A inhibition. The authors also conclude that cells sensitive to KIF18Ai have lower basal APC activity, which "drives cancer vulnerability to KIF18A inhibition." As detailed below, this point is not completely supported by the data, and in this reviewer's opinion, this conclusion (and the paper title) should be revised. Despite the somewhat expected results from the screens, the data presented do increase our understanding of the details and possible downstream effects of KIF18Ai-induced mitotic arrest in several cell types. Overall, this is a well-conducted study that will be of interest to the mitosis and cancer biology fields. Several points are outlined below that I feel should be addressed prior to publication.

Major Concerns:

A primary conclusion of the paper is that vulnerability of cancer cells to KIF18A inhibition is "driven" by an imbalance of SAC signaling and APC activity. While the authors demonstrate nicely that changes in SAC signaling or APC activity can influence cellular responses to KIF18A inhibition, the data fall short of establishing that changes to SAC signaling or APC are the bona fide drivers of vulnerability in tumor cells. The conclusions in the paper are primarily based on evidence that mitotic duration roughly correlates with sensitivity to KIF18A inhibition. This is apparent when both sensitive and insensitive cell types are plotted on the same graph (Fig 6A). However, the correlation between toxicity and mitotic duration does not look that convincing for sensitive cell lines alone. For example, MDA-MB-157 cells have one of the longest mitotic durations but the lowest toxicity of the sensitive cell lines tested. The authors also conclude that sensitive cell lines have lower basal APC activity than insensitive cell lines. However, this is based Cyclin B degradation rates in HeLa cells compared to RPE1 cells following treatment with MPS1 inhibitors (Fig 6F). The data suggest that some HeLa cells show similar degradation rates to RPE1 cells under these conditions but there is a larger variance among individual HeLa cells. Thus, it's not clear whether basal APC activity is actually that much lower in the HeLa cells or whether this trend would hold true across other sensitive cell types. It would also be important to know if CIN/ aneuploid cells that are not sensitive to KIF18A have higher APC activity or reduced SAC function. Personally, I don't think all of this needs to be worked out for the current paper but do think the conclusions should be toned-down such that they better reflect the data presented.

Given that only a few cell types were analyzed in detail in this study, the authors should be cautious about making general conclusions regarding the basis of KIF18A dependency.

To facilitate a complete interpretation of the data, more information about the KIF18A inhibitor used in this study, AM-1882, needs to be provided. The structure and validation data for the inhibitor are not presented, and the authors cite another

manuscript that is "in review."

In reference to data in Figures 1 & 3, the authors describe observing hyper-oscillation of chromosomes in HCC1806 but not HeLa cells following KIF18Ai. Based on these observations, the authors conclude that increased SAC signaling can occur in the absence of increased oscillations. However, no quantification of chromosome oscillations is provided and it is not described how the presence of hyper-oscillations were determined. In the one example movie provided for HeLa cells (Mov 5), it appears that the time-resolution may be too low to reliably assess chromosome oscillations. Therefore, the relationship between chromosome movements and SAC signaling should be more thoroughly measured or this conclusion should be removed from the manuscript.

Minor Concerns:

HeLa cells are listed as a model for ovarian cancer but were actually derived from a cervical tumor

Relevant references that should be considered:

The survival of KIF18A loss of function mice were first described in PMID 20981267 and 25824710.

Micronucleation as a result of KIF18A loss of function was first described in PMID 30733233

Referee #1:

General summary and opinion about the principal significance of the study, its questions and findings

In this manuscript, Glietch et al perform a detailed mechanistic analysis of a recently developed KIF18A inhibitor, which displays preferential toxicity in cancer lines. This work is important and timely because recent data demonstrated that cancer cells with tetraploidy or chromosomal instability (CIN) are more sensitive to KIF18A knockdown/knockout, implying that small molecule inhibitors targeting KIF18a may be useful for treating a wide range of cancer types. To facilitate effective use of these drugs in the clinic, it is crucial to understand the molecular basis for KIF18a sensitivity across cell types.

This manuscript starts by performing a thorough characterisation of a panel of cell lines (non-transformed and different cancer types) that are sensitive and insensitive to KIF18A inhibition. CRISPR screening is used to identify different resistance mechanisms, and these are independently validated for three different targets. These studies led to the conclusion that KIF18A inhibition affects chromosome oscillations and kinetochore attachments to produce a SAC dependent delay, which ultimately arrests cells in mitosis and leads to toxicity. A further CRISPR screen led to the discovery that reduced APC activity sensitizes cells to KIF18A inhibition by enhancing the SAC-dependent delay. Importantly, this appears to be a strong predictor of KIF18a sensitivity, because mitotic and metaphase duration correlated strongly with sensitivity across the panel of tested cell types, and DepMap analysis implies this relationship holds true across many more cancer lines. Finally, analysis of live cyclin B degradation assays demonstrate sensitivity is directly associated with reduced APC activity.

In summary, the authors have performed very careful and detailed mechanistic analysis, which I found to be sufficient to fully support their conclusions. I feel that the most significant conclusion is that reduced APC/C activity, which may be a common feature of cancer cells, causes sensitivity to KIF18a inhibition by magnifying the SAC-dependent delays. Another important conclusion is that elevated chromosome numbers can also magnify these delays by directly increasing the total SAC signal, and that these two features together can synergise to enhance KIF18i-sensitivity. These conclusions are particularly significant because cancer cells with CIN frequently display elevated chromosome numbers, and partial APC/C inhibition has been implicated as a common mechanism that cancer cells use to tolerate CIN. Therefore, cancer cells could well have evolved the two main features that give rise to KIF18A sensitivity. This study therefore lays the foundations to develop biomarkers that can assess these features across different cancer lines, and ultimately, patient tumours.

Specific major concerns essential to be addressed to support the conclusions

I have no major concerns

Minor concerns that should be addressed

1) I cannot find the legends that describes the details/titles of Movies S1 - S5?

These have been generated and included in the revised manuscript.

2) In relation to: "Finally, we find that the degree of SAC signaling does not correlate well with sensitivity"

- This should read "the degree of BUBR1 and MAD1 recruitment does not correlate well with sensitivity. SAC signaling encompasses much more than just the recruitment of these proteins to kinetochores (e.g. CDC20 recruitment and phosphorylation changes on MAD1, BUB1).

This point is well taken, and the phrasing has been adjusted.

3) in relation to: "these drugs have failed to progress in the clinic."

- Some are still in clinical trials, so it is perhaps better to state that they have "so far failed to progress in the clinic."

This phrasing has been modified as suggested.

4) In relation to: "However, a G289I KIF18A mutant unable to bind PP1 largely rescued growth (Figures 2H and S2B)."

- I think the referenced figure should be 2I.

This error was noticed after submission and has been corrected.

5) A very recent manuscript from Tsang and Cheeseman provides a potential explanation for differences in mitotic cell fate following a prolonged mitotic arrest (relating to difference in CDC20 variants). This could perhaps be mentioned in the relevant section of the discussion.

This paper was mentioned by several reviewers which encouraged us to generate and test the effect of CDC20 WT and M43 translational isoform variant overexpression on KIF18Ai toxicity. Consistent with our finding that weakened APC/C promotes KIF18A dependency, overexpression of these proteins which is predicted to activate the APC/C relative to the SAC, dramatically reduced KIF18Ai toxicity. This is detailed in Figure 6H across the panel of sensitive cell lines and is has been added to the discussion.

Any additional non-essential suggestions for improving the study (which will be at the author's/editor's discretion)

It is interesting that KT-MT turnover is more strongly affected by KIF18 inhibition in sensitive HCC1806, in comparison to insensitive RPE1 cells (Fig.3I). Is this a common feature that could also explain KIF18a sensitivity across more cell lines? It may be challenging to performed detail photo-activation experiments in multiple cell types, but the effects of rapid destabilisation of KT-MT with cold-shock +/- KIF18A inhibition may reveal important differences in sensitive and insensitive cell types.

As suggested, we conducted the KT-MT cold-shock assay in HCC1806 cells. We readily detected bulk microtubule stabilization by Taxol in 30 min Cold shock and 6 min Nocodazole shock (Reviewer Figure A), and with low-dose Taxol treatments (Reviewer Figure B). However, we found cold/nocodazole shock assays did not have the sensitivity to detect the changes in KT-MT stability induced by KIF18Ai. Even with a shorter depolymerization step to better capture KIF18Ai destabilization (20 min Cold Shock, Reviewer Figure C), the slight downward trend in microtubule stability seen with KIF18Ai treatment was far from reaching statistical significance over three independent experiments. These findings are consistent with the evaluation of assays for measuring kinetochore-microtubule attachments conducted by Warren JD et al 2020, (PMID 32423652) which showed the photo-activatable GFP-Tubulin assay to be more sensitive at detecting small effects in KT-MT stability.

Since cold shock assays didn't provide the required sensitivity for our experiments, we expanded our PA-GFP-Tubulin analysis to the bulk of the cell line panel (Figures 2I and S7D-E). Unfortunately, we were unable to generate MDA-MB-157 cells with this system. We found that, while baseline K-MT attachment stability was highly variable, the degree of disruption of K-MT attachments caused by KIF18Ai was weakly predictive of KIF18Ai toxicity (Figure 3J). We conclude that a heightened reliance on KIF18A for K-MT stabilization contributes to, but is not fully predictive of, toxicity in KIF18Ai.

In relation to the above, I appreciate that SAC strength/duration of mitosis is an important predictor of outcome, but should the direct sensitivity to KT-MT destabilisation following KIF18i not also be highlighted as a potential mechanism of sensitivity? It certainly tracks with HCC1806 sensitivity (-/+ HSET knockout), so is there a reason to believe this will not also be the case in other sensitive lines? I understand that cell lines that are sensitive for this reason would be predicted to also display higher BUBR1/MAD1 signals, but could this explain sensitivity in MDA-MB-157 cells, for example, which do have high BUBR1/MAD1?

Since SAC activation at individual kinetochores is downstream of K-MT attachment stability, we originally treated this as one sensitization mechanism. However, this point is well-taken, and in our revised manuscript we focused more directly on the variable reliance on KIF18A for K-MT stabilization as a factor that may drive a SAC:APC/C imbalance. This has been incorporated in the results section as well as in our discussion.

Adrian T Saurin

Referee #2:

Accurate chromosome segregation requires the successful attachment of chromosomes via their kinetochores to the mitotic spindle. This process is modulated by a range of enzyme and non-enzyme proteins localized both to spindle microtubules and kinetochores. Recently, the mitotic motor protein KIF18A, necessary for normal chromosome alignment during mitosis, has attracted a lot of attention because it was found to be specifically required for successful cell division in cells with abnormal ploidies, suggesting that it may constitute a useful target in the therapy of tumours characterised by aneuploidy. However, despite these important findings, it has so far remained elusive which molecular differences determine the differential cell viability upon loss of KIF18A.

In this manuscript Glietch and colleagues analyse the response of a panel of non-transformed and transformed cell lines to a recently obtained small molecule inhibitor of KIF18A (referred to as KIF18Ai). Informed by cell viability data in the presence or absence of the drug, the authors split their panel of cells into KIF18Ai-sensitive and insensitive cell lines and then try to identify key factors for the differential outcome. Using a combination of CRISPR/Cas9 knockout screens in KIF18Ai-sensitive and partially sensitive cell lines and cell biological analysis of microtubule-kinetochore attachment formation, spindle assembly checkpoint function and activity of the anaphase promoting complex/cyclosome (APC/C) the authors come to the conclusion that the determining factor for KIF18Ai-sensitivity is the precise balance of spindle checkpoint signalling versus anaphase promoting complex/cyclosome (APC/C) activity in different cell lines.

The question of why loss or inhibition of KIF18A is toxic in some cell lines but not in others is very interesting. While this manuscript attempts to uncover the molecular biology governing the response to loss of KIF18A, the results are not particularly enlightening. This is partially due to the dense and confusing lay-out of the paper which presents a large number of experiments looking at different aspects of chromosome segregation process in genetically different cell lines without a clear rationale of why some experiments are conducted in one cell line while others are carried out in a different cell line, and without providing real depth to any single point of investigation.

We regret that our data presentation was found to be dense and hard to interpret. Wherever possible, we attempted to use unbiased quantitative metrics to enable the rigorous analysis of the biology seen in immunofluorescence images and live-cell timelapse experiments. By nature, this generated data-dense figures that we did our best to make accessible to the reader. Nevertheless, where specifically noted, we have attempted to reinforce our quantitative analysis with a greater number of representative images, as well as tried to simplify our data representation overall.

In response to the point about the depth of our study, our goal was to outline the large-scale determinants of KIF18A dependency. Unlike the effect of increases in chromosome number, both SAC hyperactivation and APC/C deficiency may have complex underlying molecular mechanisms which are likely divergent in our panel of cell lines (outlined in our discussion). This explains the lack of expression or mutational signatures associated with KIF18Ai sensitivity. As a result, we favored outlining high-level, common phenotypes, such as slow mitosis/metaphase to anaphase transition among cell lines rather than attempting to determine the specific molecular cause of each deficiency in each sensitive cell. We acknowledge that there remain several interesting outstanding details to elucidate such as what dictates the divergent cellular responses to mitotic delay, and we have attempted to outline these areas in our revised discussion. Nevertheless, we believe that we have uncovered a novel generalizable rule set that

determines KIF18A dependency, and our work will catalyze future studies of this important cancer target.

The cell lines used in our manuscript were chosen due to reagent availability and KIF18Ai sensitivity following a ~900 cell line PRISM drug screen run in the Payton et al publication attached to our submission (now accepted at Nature Cancer). Whenever possible, we assayed all 10 cell lines, but for some experiments that required genetic alterations (e.g. endogenous Cyclin B1 tagging), we were unable to generate these tool cell lines across the entire panel as not all our cell lines were amenable to genetic modification and single-cell cloning. We have, however, expanded several key analyses to a larger number of cell lines, which are specifically noted below.

Most of the data are buried in difficult to interpret graphs, preventing the reader to assess the quality of the raw data. The constant change between cell lines is extremely confusing. The manuscript contains many quite general statements (e.g. "We conclude that hyperploidy and low APC/C activity act synergistically to generate KIF18A dependency.") when the data were only obtained with only one of the cell lines (in this example RPE1 cells).

This point is well-taken. We have expanded the experiments with endogenous tagging of Cyclin B1 to a greater number of cell lines. Consistent with our original conclusions, anaphase duration and Cyclin B1 degradation is slower in sensitive lines. Interestingly, these latest experiments revealed that Cyclin B1 degradation is slower either because of weak baseline APC/C activity or weak APC/C as a result of persistent SAC signaling through the metaphase to anaphase transition (Figure 6F-G and S10B-C). This feature is now noted in both the results and discussion sections and adds additional mechanistic detail to how KIF18A dependency is derived.

Given that the authors set out to identify more general determinants of KIF18Ai sensitivity, their experiments should be repeated with the entire panel of cell lines before such statements are made.

In relation to changes in cell lines, we aimed to run all critical experiments across all cell lines in the panel, notably: mitotic response to KIF18Ai (Fig 1E), the rescue of toxicity with SAC inhibition (Fig 2E), SAC activation in KIF18Ai (Fig 4A-D), Ploidy relative to KIF18Ai toxicity (Fig 4G), and several APC/C correlates with toxicity (Fig. 6A-D). Furthermore, we ran our CRISPR/Cas9 screens across three independent sensitive cell lines so that we could best generalize our findings.

Nevertheless, we agree that several experiments would benefit from the inclusion of additional cell lines when possible, which we have addressed as follows:

APC/C activity and Cyclin B1 degradation: Using a recently published RNP nucleofection approach (noted in methods) we were able to generate endogenous tagging of Cyclin B1 in four additional cell lines: DLD1, HCT116, HCC1806, and OVCAR-8. These additional cell lines further reinforced our finding that KIF18Ai-sensitive cell lines have a naturally weak effective APC/C activity (Figure 6F-G and S10C-D). As noted above, we were excited to uncover that persistent metaphase SAC was another mechanism by which some of these cells were delaying the metaphase to anaphase transition and increasing KIF18A dependency.

Kinetochores-microtubule stability measurements: Variable reliance on KIF18A for kinetochores-microtubule stability is an obvious candidate for differential sensitivity between cell lines. As a result, we expanded our PA-GFP-Tubulin analysis to the bulk of the cell line panel (Figures 2I and S7D-E).

Unfortunately, we were unable to generate MDA-MB-157 cells with this system. We found that, while baseline K-MT attachment stability was highly variable, the degree of disruption of K-MT attachments caused by KIF18Ai was weakly predictive of KIF18Ai toxicity (Figure 3J). We conclude that a heightened reliance on KIF18A for K-MT stabilization contributes to, but is not fully predictive of, toxicity in KIF18Ai.

There is also a degree of uncertainty as to what the authors are actually investigating. In their first experiments they look at cell viability 5 days after exposure of cells to KIF18Ai. There have recently been several publications (some from the authors' own lab) demonstrating that perturbed mitosis can lead to p53-dependent cell cycle arrest or cell death in the following G1 phase. The p53 status of the cell lines the authors are investigating is therefore very important for the interpretation of the results and should be disclosed in the form of a table in Figure 1.

We appreciate this point and have now included the p53 status in Figure 1 C and have more clearly summarized cell line features in Dataset EV1. The p53 status was assessed by testing the sensitivity of the cell line panel to the MDM2 inhibitor Nutlin-3 (Figure S1D), as only cells retaining p53 function can arrest in Nutlin-3. Interestingly, all of the KIF18Ai-sensitive cell lines are p53 deficient. This trend of p53 loss in KIF18A-dependent cancers was also noted in Payton et al. Nat. Cancer 2023 (included with this resubmission) and is thought to be because p53 prevents the proliferation of cells with whole genome doubling or chromosomal instability – two common features of KIF18Ai-dependent lines. To directly test the effect of p53 function on KIF18A dependency, we performed 5-day MTT assays in 2N and 4N RPE1 p53-knockout cells. We found that, unlike APC/C activity, p53 status had no direct effect of KIF18Ai sensitivity (Reviewer Figure D).

In the rest of the manuscript, the authors focus mainly on the direct consequences of KIF18Ai for mitotic progression without mentioning or discussing that the loss of cell viability may be an indirect effect of the cell-line specific response to mitotic perturbations in the next G1 phase, not the mitotic perturbation in itself.

The fact that p53 function is preferentially lost in sensitive cell lines argues against this interpretation as p53-deficient cells are likely to be less sensitive to mitotic errors in the following cell cycle. Nevertheless, we agree that cells have highly variable responses to mitotic delays and that this variability likely affects overall toxicity (i.e. chromosome mis-segregation is less lethal than mitotic cell death). However, unlike the propensity for mitotic delay, the nature of mitotic errors was not well-correlated with KIF18Ai dependency (Figures 1E and S3A), and the fact that most mitotic events in sensitive cell lines result in errors suggests an untenable accumulation of defects in the population over several generations. Therefore, while we agree that an intolerable level of aneuploidy following DNA segregation errors is a likely mechanism for toxicity, since *all types* of mitotic errors are tightly correlated with mitotic delays in KIF18Ai, we feel it is most appropriate to treat mitotic delay as the 'upstream' driver of toxicity.

We also agree that why a cell line may see 75% vs. 100% toxicity in KIF18Ai is likely influenced by a variable response to mitotic delay (in mitosis *and* the following interphase). However, this phenomenon is beyond the scope of our manuscript and is, therefore, mentioned as an opportunity for future research in our discussion. This is especially interesting in the context of MCF7 cells, which are delayed in KIF18Ai but do not exhibit high rates of mitotic errors.

This is particularly surprising because their own data show that the mitotic response of the cell lines does not seem to be the differentiating factor ("Unexpectedly, we also found that both sensitive and

insensitive cell lines experienced similar increases in the per-kinetochore BUBR1 signal and the number of MAD1+ kinetochores per cell in response to KIF18Ai (Figures 4A-D). Ultimately, this reveals that SAC signaling at kinetochores is a poor overall predictor of KIF18Ai sensitivity (Figures 4B and 4D)."). In other words, are p53 (functionally) negative HeLa cells sensitive to KIF18Ai because they undergo defective mitosis and don't stop cell proliferation in response to this and then die, whereas p53 wild type RPE1 cells enter a prolonged G1 arrest in response to minor mitotic disturbances caused by KIF18Ai and hence survive?

We agree that continued proliferation in the context of p53 deficiency is likely leading to an overwhelming burden of mitotic errors in HeLa cells. However, RPE1 cells fail to delay in mitosis or experience large-scale mitotic errors (Figure 1E and S3A) despite having similar levels of SAC activation in KIF18Ai. Moreover, loss of p53 did not reduce the sensitivity of 2N or 4N RPE1 cells to KIF18Ai (Reviewer Figure D), and there is little evidence that the insensitive cell lines are eliminating large amounts of problematic cells, which would be seen as a significant slowing in growth potential (such as in Figure 5H, RPE1 2N). This argues that the incidence of errors, not the p53-dependent response to errors in interphase, is the primary driver of KIF18Ai toxicity.

There are some live cell imaging experiments of cells exposed to KIF18Ai (e.g. Figure 1E), which may be interesting in this regard, but from the way the data are presented it is difficult to find out how many of the cells stop dividing during the course of the experiments.

We only analyzed the first mitotic event following KIF18Ai treatment and have no longitudinal analysis in our live-cell timelapse movies following mitotic exit due to the timeframe of imaging. Tracking individual cells by time-lapse microscopy over 5 days is very challenging. However, the high burden of mitotic errors seen in sensitive cell lines appears sufficient to explain the longitudinal toxicity seen in the 5-day MTT assay.

However, from the numbers in Supplemental table 1 it seems that far fewer mitotic events were recorded in RPE1 cells after exposure to KIF18Ai than without, in line with the idea that these cells stopped proliferating.

The number of cells analyzed was due to confluency and field-of-view constraints in the videos and is not a proxy for cell growth. Instead, we can use the 5-day MTT assays, which carefully assess cell growth and viability. In addition, confluency measurement on the CellCyte-X (Fig 5H) shows that the growth of RPE1 cells over 7 days is unperturbed by KIF18Ai treatment.

In summary, in the current form, the manuscript does not clearly differentiate between direct mitotic defects incurred by KIF18Ai and the cellular response to defective mitosis, and despite the authors' goal of defining the molecular biology that render cells sensitive to KIF18Ai generally, the results that the authors report appear to be specific to certain cell lines.

The point concerning the roll out of critical assays to the full panel is appreciated. As indicated above, whenever possible, we made our best effort to expand our analysis to all 10 cell lines in the panel, which have divergent genetic and tissue backgrounds. Furthermore, the contribution of weak effective APC/C activity to sensitivity, a finding unique to this study, was further supported by the Broad DepMap dataset comprising over 600 cell lines (Figures 6D and S10A). We therefore feel it is appropriate to generalize some of our claims. Nevertheless, specific comments related to the generalization of conclusions were

also made by other reviewers and are well taken. We have adjusted our language surrounding experiments that do not use the full panel of cell lines to properly reflect their contribution to our model.

Specific comments:

- **Figure 1:** The reason for the choice of cell lines should be more clearly articulated, and a table should be provided to specify p53 status, ploidy and origin of the chosen cell lines. HeLa is not an ovarian cancer!! Since ploidy has previously been described as a differentiating factor for sensitivity or insensitivity to KIF18A depletion it is particularly important that the cell lines are classified according to this. How do the authors findings tie in with the idea that ploidy is the differentiating factor for sensitivity to loss of KIF18A?

We appreciate this reviewers request for additional clarity on this point. We have now included in our manuscript the fact that the cell lines used in our manuscript were chosen due to reagent availability and KIF18Ai sensitivity following a ~900 cell line PRISM drug screen run in the Payton et al publication attached to our submission (now accepted at Nature Cancer). We have also included a new supplemental table summarizing this and other useful cell line information (Dataset EV1).

Furthermore, we have extensively outlined the contribution of ploidy to KIF18A dependency throughout the manuscript, notably in Figures 4E-G and S8 which conclude that higher ploidy amplifies the KIF18Ai-driven SAC signal across a larger number of kinetochores resulting in a larger cumulative SAC burden at metaphase. In agreement with other recent work, we find hyper-ploidy to be a key feature driving KIF18A dependency which we have outlined in our model in Figure 7.

- **It is extremely confusing that not all of the experiments are conducted in all cell lines, and there is no proper explanation of why some experiments are conducted in one cell line and not another. Given that the authors argue that their results may be important clinically, it is critical that their conclusions are checked in all of their cell lines, and that the genetic background (transformed/untransformed, p53 status, ploidy etc.) is taken into consideration and discussed when the results are interpreted.**

We attempted to run all critical experiments across the full cell line panel (as listed above), however some experiments required complicated genetic manipulation and/or clonal isolation which was not possible for us in some cancer cell lines. Nevertheless, renewed attempts have enabled us to expand several critical experiments (photoactivatable-GFP-tubulin and Cyclin B1 endogenous tagged lines) to a greater number of cell lines.

As a fallback we deferred to HCC1806 as a sensitive cell line model since they were highly sensitivity to KIF18Ai and shared the most common features with other sensitive cell lines (CIN, p53 defective, slightly hyperploid, long mitotic delays with a mix of cell death and problematic mitotic exit). By contrast, RPE1 cells were used as the model insensitive line as they reasonably represent non-transformed cells. We have additionally changed conclusions when appropriate to reflect whether they were drawn from a single cell line or the full cell line panel.

- **Figure 1A:** The authors state "Treatment with KIF18Ai causes stable binding of the KIF18A motor domain to microtubules (Tamayo et al., 2022), leading to the relocation of KIF18A in mitosis from kinetochores to spindle poles in all the cell lines tested (Figures 1A and S1A). This mis-localization phenocopies the effect of KIF18A loss (Payton et al., 2023), likely by blocking KIF18A-induced

stabilization of microtubule plus ends(Du et al., 2010)." This statement does not make any sense - surely in cells without KIF18A, there cannot be mis-localisation of KIF18A? Also, the referenced paper "Payton et al, 2023" is under review according to the reference list (and does not appear to be on bioRxiv) so cannot be cited as evidence, as the reader has no access to this information.

Thank you for highlighting this confusing phrasing. KIF18A stabilizes microtubule plus ends at the metaphase plate. Since KIF18A protein is being mis-localized to spindle poles after inhibitor treatment, it cannot reach microtubule plus ends to exert its stabilizing effect. This equates to the loss of KIF18A function at the kinetochore and phenocopies KIF18A knockout. The phrasing has been adjusted to avoid confusion.

Toxicity of KIF18A loss and KIF18A inhibition is well correlated across cell lines which is seen in the Payton et al 2023 paper. This manuscript was just accepted in Nature Cancer, and we have provided the final accepted version with this resubmission.

- Figure 3: "Knockout of HSET and, to a lesser extent Cyclin B1, dramatically improved the ability to form and maintain a metaphase plate in KIF18Ai (Figure 3B)." This information is not visible in Figure 3B. Could the authors please show representative time lapse images instead of the very confusing graph? The information that is easily extracted from the graph is that mitotic duration is drastically reduced in cells lacking cyclin B1 but that is much more likely due to an abrogation of mitosis altogether rather than an improvement of the cells' ability to hold a metaphase plate. Notably, this graph lacks a WT control.

We have attempted to improve the clarity of this panel based off this comment. Furthermore, representative stills from the movies can now be found in Figure S6A. Figure 3B summarizes the cell's ability to maintain a metaphase plate as it reports the longest continuous metaphase for each cell line trace. We believe therefore that HSET loss counteracts the effect of KIF18Ai treatment by dampening chromosome alignment defects which in turn favors mitotic exit. WT control is shown in the two columns on the far left.

- Figure 3: The cell biological analysis presented in the manuscript relating to the stability of microtubule-kinetochore attachments needs to be improved. The only experiment that was conducted to demonstrate a defect in microtubule-kinetochore attachment stability upon KIF18Ai is the decay of photo-activatable GFP-alpha-tubulin (Figure 4). This was only done in one "insensitive" cell line (RPE1) and one "sensitive" cell line (HCC1806) so does not give any information about whether any differences are reproducible for the other insensitive or sensitive cell lines. Furthermore, this assay is difficult to interpret because of the inherently different rates of microtubule flux in the different cell lines. A much more informative assay would be the classical cold-stability assay combined with the measurement of inter-kinetochore stretch. As these assays do not rely on genetic modifications of the cell line to be investigated, they can easily be done on a panel of different cell lines and would hence lead to comparable results. As it stands, the statement that "Taken together, our data show that treatment with KIF18Ai leads to a weakening of K-MT attachments"" is not sufficiently supported by data.

We conducted the KT-MT cold-shock assay in HCC1806 cells. We readily detected bulk microtubule stabilization by Taxol in 30 min Cold shock and 6 min Nocodazole shock (Reviewer Figure A), and with low-dose Taxol treatments (Reviewer Figure B). However, we found cold/nocodazole shock assays did not have the sensitivity to detect the changes in KT-MT stability induced by KIF18Ai. Even with a shorter depolymerization step to better capture KIF18Ai destabilization (20 min Cold Shock, Reviewer Figure C),

the slight downward trend in microtubule stability seen with KIF18Ai treatment was far from reaching statistical significance over three independent experiments. These findings are consistent with the evaluation of assays for measuring kinetochore-microtubule attachments conducted by Warren JD et al 2020, (PMID 32423652) which found the photo-activatable GFP-Tubulin assay to be more sensitive at detecting small effects in KT-MT stability. Similarly, we have attempted several times to measure inter-kinetochore stretch in our cell line panel and were ultimately unsuccessful.

We agree that our analysis of KT-MT stability could be improved by investigating a larger number of cell lines. We have therefore expanded our PA-GFP-Tubulin analysis to the bulk of the cell line panel (Figures 2I and S7D-E). Unfortunately, we were unable to generate MDA-MB-157 cells with this system. We found that, while baseline K-MT attachment stability was highly variable, the degree of disruption of K-MT attachments caused by KIF18Ai was weakly predictive of KIF18Ai toxicity (Figure 3J). Given the expansion of our dataset, we also now find it appropriate to claim that KIF18A broadly destabilizes K-MT attachments. Furthermore, we conclude that a heightened reliance on KIF18A for K-MT stabilization contributes to, but is not fully predictive of, toxicity in KIF18Ai.

• Figure 6: In this figure, the authors conclude from their cyclin B degradation assays that HeLa cells have inherently lower APC/C activity than RPE1 cells and that this is the reason that HeLa cells are sensitive to KIF18Ai. The authors then go on to test this hypothesis by overexpressing the ubiquitin E2 ligase UBE2S to increase APC/C activity, but surprisingly and nonsensically, do not carry out this experiment in HeLa cells but in HCC1806 cells. Why was the experiment not performed in HeLa cells (or indeed all cell lines of their panel), and could this experiment please be conducted?

We have expanded our UBE2S overexpression analysis to our three most sensitive cell lines (HCC1806, HeLa, and OVCAR-3). Interestingly, KIF18Ai viability rescue was only seen in HCC1806 and HeLa cells which suggests that either UBE2S is not rate-limiting for APC/C activity in OVCAR-3 cells or that APC/C hyperactivity is insufficient to overcome the KIF18Ai-generated SAC signaling in these cells (Figure 6I). To bolster this line of reasoning further however, we have now additionally generated APC/C hyperactivity through the overexpression of CDC20 or a truncated CDC20 M43 translational isoform (from PMID 37100900). Consistent with the UBE2S data, these overexpression cell lines saw significant decreases in KIF18Ai toxicity relative to WT controls across the full sensitive line panel (Figure 6H).

Referee #3:

This manuscript from Gliech and colleagues investigates the molecular determinants of cancer cell dependency on the kinesin KIF18A. Previous screens identified KIF18A as being uniquely essential in a subset of cancer cells. This prior work generated interest in KIF18A as a target for cancer therapy, and several pharmaceutical companies (including Amgen- which employs authors on this paper) have announced programs focused on developing and testing anti-KIF18A inhibitors. However, what the underlying differences are that confer sensitivity to KIF18A in some cancer cells remains an important, unanswered question.

To address this, the authors performed whole genome CRISPR screens to identify genes that enhance or suppress the effects of a KIF18A inhibitor on proliferation of HCC1806 and OVCAR-8 cells. The top hits from the screens have roles in the SAC and regulation of microtubules. The authors provide detailed characterizations of cellular phenotypes following combined KIF18A inhibition and knockout/ knockdown of MAD1, HSET, or Cyclin-B. The data presented are impressive and thorough. However, the conclusions are not particularly surprising in the context of prior work though. For example, previous studies have demonstrated that the mitotic arrest and loss of viability seen after inhibition of KIF18A function depend on the SAC. Given this information, and the known relationship between the SAC and the APC, it is also not surprising that inhibition of the APC enhances the arrest caused by KIF18A inhibition. The authors also conclude that cells sensitive to KIF18Ai have lower basal APC activity, which "drives cancer vulnerability to KIF18A inhibition." As detailed below, this point is not completely supported by the data, and in this reviewer's opinion, this conclusion (and the paper title) should be revised. Despite the somewhat expected results from the screens, the data presented do increase our understanding of the details and possible downstream effects of KIF18Ai-induced mitotic arrest in several cell types. Overall, this is a well-conducted study that will be of interest to the mitosis and cancer biology fields. Several points are outlined below that I feel should be addressed prior to publication.

Major Concerns:

A primary conclusion of the paper is that vulnerability of cancer cells to KIF18A inhibition is "driven" by an imbalance of SAC signaling and APC activity. While the authors demonstrate nicely that changes in SAC signaling or APC activity can influence cellular responses to KIF18A inhibition, the data fall short of establishing that changes to SAC signaling or APC are the bona fide drivers of vulnerability in tumor cells. The conclusions in the paper are primarily based on evidence that mitotic duration roughly correlates with sensitivity to KIF18A inhibition. This is apparent when both sensitive and insensitive cell types are plotted on the same graph (Fig 6A). However, the correlation between toxicity and mitotic duration does not look that convincing for sensitive cell lines alone. For example, MDA-MB-157 cells have one of the longest mitotic durations but the lowest toxicity of the sensitive cell lines tested. The authors also conclude that sensitive cell lines have lower basal APC activity than insensitive cell lines. However, this is based Cyclin B degradation rates in HeLa cells compared to RPE1 cells following treatment with MPS1 inhibitors (Fig 6F). The data suggest that some HeLa cells show similar degradation rates to RPE1 cells

under these conditions but there is a larger variance among individual HeLa cells. Thus, it's not clear whether basal APC activity is actually that much lower in the HeLa cells or whether this trend would hold true across other sensitive cell types. It would also be important to know if CIN/ aneuploid cells that are not sensitive to KIF18A have higher APC activity or reduced SAC function. Personally, I don't think all of this needs to be worked out for the current paper but do think the conclusions should be toned-down such that they better reflect the data presented.

We appreciate the limitations of studying this phenomenon across a small panel of cell lines. We have addressed this point in several ways. Firstly, we demonstrated through metaphase to anaphase duration measurements that all sensitive cell lines were APC/C activity deficient at mitotic exit. Furthermore, using the large (>600 cell line) DepMap dataset, we established a wide-reaching correlation between sensitivity to loss of APC/C activity and KIF18A loss which is strongly indicative of an issue with APC/C activity at the metaphase to anaphase transition.

In this resubmission, we have also been able to expand direct analysis of APC/C function to four additional cell lines now shown in Figure 6F-G and S10C-D. Interestingly, we found that low apparent APC/C activity can both be the result of a combination of weak basal APC/C activity *and* SAC signal persisting through the metaphase to anaphase transition. Variability in apparent APC/C activity is likely due to the degree of persistent metaphase SAC contribution, which is stochastic from mitosis to mitosis within each cell line. Nevertheless, weak *effective* APC/C activity at mitotic exit correlates strongly with KIF18Ai toxicity. This change is now reflected in a changed title for the manuscript.

Our cell line panel was not designed to specifically include CIN/aneuploid cells in both the sensitive and insensitive groups, so we cannot draw conclusions in that regard. CIN and aneuploidy scores (as well as additional information) have now been included in Dataset EV1. However, maintaining CIN while changing SAC:APC/C activity was addressed in several experiments in which we overexpressed CDC20 or UBE2S to generate rescue in CIN cell lines (Figures 6H-1).

Given that only a few cell types were analyzed in detail in this study, the authors should be cautious about making general conclusions regarding the basis of KIF18A dependency.

We have adjusted certain claims that were made based off subsets of the cell line panel. However, we maintain that an underlying imbalance between SAC and APC/C at the metaphase to anaphase transition is a broad correlate for sensitivity based off the experiments run across the entire cell line panel as well as the larger DepMap dataset analysis. This now includes a much broader analysis of K-MT attachment stability with KIF18Ai as well as an expansion of the number of cells in which Cyclin B1 dynamics were measured at mitotic exit.

To facilitate a complete interpretation of the data, more information about the KIF18A inhibitor used in this study, AM-1882, needs to be provided. The structure and validation data for the inhibitor are not presented, and the authors cite another manuscript that is "in review."

This manuscript was just accepted in Nature Cancer, and is provided to the reviewers with the resubmitted version of our manuscript.

In reference to data in Figures 1 & 3, the authors describe observing hyper-oscillation of chromosomes in HCC1806 but not HeLa cells following KIF18Ai. Based on these observations, the authors conclude that increased SAC signaling can occur in the absence of increased oscillations. However, no quantification of

chromosome oscillations is provided and it is not described how the presence of hyper-oscillations were determined. In the one example movie provided for HeLa cells (Mov 5), it appears that the time-resolution may be too low to reliably assess chromosome oscillations. Therefore, the relationship between chromosome movements and SAC signaling should be more thoroughly measured or this conclusion should be removed from the manuscript.

It appears that we are using slightly different definitions for “chromosome oscillation” which has caused an issue with clarity. Our analysis was conducted with 5 min time resolution and therefore could not follow individual chromosomes as they oscillated in and out of the chromosome mass. However, the presence of chromosomes closer to the poles that dynamically appear and disappear as they re-enter the metaphase plate suggests that these chromosomes are either rapidly attaching and detaching from the spindle, or what seemed more likely, attached but oscillating. This is also consistent with previous KIF18A literature describing this phenomenon. We did not use a specific distance measurement from the poles/plate for what constitutes “oscillating” vs “congressed”, rather we analyzed individual frames and compared them to a series of reference images (defined in the figure legend) as a means of broadly quantifying chromosome behavior. We have increased the level of detail describing this experiment in the results section as well as material and methods.

With regards to whether HSET knockout predominantly rescues growth through metaphase stability (reducing chromosome missegregation errors at mitotic exit), or K-MT attachment stability (transient SAC silencing) is hard to determine as these two things are functionally intertwined. We can change our language however to better represent this nuance.

Minor Concerns:

HeLa cells are listed as a model for ovarian cancer but were actually derived from a cervical tumor

We appreciate this reviewer highlighting this error. It has now been corrected in the text.

Relevant references that should be considered:

The survival of KIF18A loss of function mice were first described in PMID 20981267 and 25824710.

Micronucleation as a result of KIF18A loss of function was first described in PMID 30733233

We appreciate this being brought to our attention. The appropriate references have been added.

Dr. Andrew J Holland
Johns Hopkins School of Medicine
Dept of Molecular Biology & Genetics
725 N Wolfe St, 704 PCTB
Baltimore, 21205

14th Dec 2023

Re: EMBOJ-2023-114411R
Weakened APC/C activity at mitotic exit drives cancer vulnerability to KIF18A inhibition

Dear Andrew,

Thank you again for submitting your revised manuscript to The EMBO Journal. We have now finally received re-reviews from original referees 2 and 3 (copied below), and I am happy to say that both of them are overall satisfied with your responses and revisions. Following a final round of minor revision to incorporate the remaining textual suggestions of referee 3, we should therefore be ready to accept the study for publication.

In addition, there are a number of editorial issues that need to be addressed at this stage:

- First of all, we still need you to complete (and upload) our dedicated author checklist, which you should download via the link in the detailed instruction section below.
- Please refer to our author guide (www.embopress.org/page/journal/14602075/authorguide#expandedview) regarding "supplementary material", and consider re-organizing the current figures and supplemental figures. We are not limited to 7 main figures (e.g. 8 or 9 would be equally fine); in addition, we can have up to 5 "Expanded View" figures (=second level), whose legends would also need to be in the main text, and which would be type-set and directly visible (expandable) with the HTML version of the paper - these figure should be named and references as "Figure EV1-5". Any additional data figures (third level below main and EB figures) should be included in a single Appendix PDF, with their respective legends below each figure, and named/referenced as "Appendix Figure S1/2/3". The Appendix should furthermore be prefaced by a brief Table of Contents, listing the included figures and their respective page numbers.
- Please also rename all movies into "Expanded View movies", adjusting the respective in-text callouts to "Movie EV1/2/3...", removing movie legends from the main text file into separate legend text files, and placing each movie together with its respective legend text file into a separate ZIP file before re-uploading.
- Since Figure 7 has only a single panel, please remove the panel label "A" both in the figure file and in the text/legends.
- Please update the reference to the accepted Payton et al article with a detailed citation, or (if not yet available) with its prospective DOI.
- As we are switching from a free-text author contribution statement towards a more formal statement based on Contributor Role Taxonomy (CRediT) terms, please remove the present Author Contribution section and instead specify each author's contribution(s) directly in the Author Information page of our submission system during upload of the final manuscript. See <https://casrai.org/credit/> for more information.
- Please provide suggestions for a short 'blurb' text prefacing and summing up the conceptual aspect of the study in two sentences (max. 250 characters), followed by 3-5 one-sentence 'bullet points' with brief factual statements of key results of the paper; they will form the basis of an editor-written 'Synopsis' accompanying the online version of the article. You may also upload a synopsis image, which can be used as a "visual title" for the synopsis section of your paper. The image (maybe simply based on Figure 7) should be in PNG or JPG format, and please make sure that it remains in the modest dimensions of (exactly) 550 pixels wide and 300-600 pixels high.
- Finally, during routine pre-acceptance checks, our data editors have raised the following queries regarding figures, data, and legends:
Please note that in (current) supplementary figure 11, legend for figure panel C is incorrectly labeled as A. This needs to be rectified.
Please define the annotated p values ****/****/**/ in the legend of figure 2c, d, e, g, i; 3b, h, i; 4a, c, e, f; 5d, f; 6h, i; supplementary figures 4d-e; 6c; 7b; 8b-c; 9e-f; 11b as appropriate.
Please indicate the statistical test used for data analysis in the legend of figure 5b.
Please note that information related to n (number of replicates) is missing in the legends of figures 1e; 2d, f, g; 3b, h, i; 4a, c, e;

6b, g; supplementary figure 10c

Please note that the error bars are not defined in the legends of figures 3h, i; 4c, e; 6g; supplementary figures 8b-c; 10c
Please note that the box plots need to be defined in terms of minima, maxima, centre, bounds of box and whiskers, and percentile in the legend of figure 3b.

I am therefore returning the manuscript to you for a final round of minor revision, to allow you to make these adjustments and upload all modified files. Once we will have received them, we should be ready to swiftly proceed with formal acceptance and production of the manuscript.

With kind regards,

Hartmut

- 1) Every manuscript requires a Data Availability section (even if only stating that no deposited datasets are included). Primary datasets or computer code produced in the current study have to be deposited in appropriate public repositories prior to resubmission, and reviewer access details provided in case that public access is not yet allowed. Further information: embopress.org/page/journal/14602075/authorguide#dataavailability
- 2) Each figure legend must specify
 - size of the scale bars that are mandatory for all micrograph panels
 - the statistical test used to generate error bars and P-values
 - the type error bars (e.g., S.E.M., S.D.)
 - the number (n) and nature (biological or technical replicate) of independent experiments underlying each data point
 - Figures may not include error bars for experiments with $n < 3$; scatter plots showing individual data points should be used instead.
- 3) Revised manuscript text (including main tables, and figure legends for main and EV figures) has to be submitted as editable text file (e.g., .docx format). We encourage highlighting of changes (e.g., via text color) for the referees' reference.
- 4) Each main and each Expanded View (EV) figure should be uploaded as individual production-quality files (preferably in .eps, .tif, .jpg formats). For suggestions on figure preparation/layout, please refer to our Figure Preparation Guidelines: <http://bit.ly/EMBOPressFigurePreparationGuideline>
- 5) Point-by-point response letters should include the original referee comments in full together with your detailed responses to them (and to specific editor requests if applicable), and also be uploaded as editable (e.g., .docx) text files.
- 6) Please complete our Author Checklist, and make sure that information entered into the checklist is also reflected in the manuscript; the checklist will be available to readers as part of the Review Process File. A download link is found at the top of our Guide to Authors: embopress.org/page/journal/14602075/authorguide
- 7) All authors listed as (co-)corresponding need to deposit, in their respective author profiles in our submission system, a unique ORCID identifier linked to their name. Please see our Guide to Authors for detailed instructions.
- 8) Please note that supplementary information at EMBO Press has been superseded by the 'Expanded View' for inclusion of additional figures, tables, movies or datasets; with up to five EV Figures being typeset and directly accessible in the HTML version of the article. For details and guidance, please refer to: embopress.org/page/journal/14602075/authorguide#expandedview
- 9) Digital image enhancement is acceptable practice, as long as it accurately represents the original data and conforms to community standards. If a figure has been subjected to significant electronic manipulation, this must be clearly noted in the figure legend and/or the 'Materials and Methods' section. The editors reserve the right to request original versions of figures and the original images that were used to assemble the figure. Finally, we generally encourage uploading of numerical as well as gel/blot image source data; for details see: embopress.org/page/journal/14602075/authorguide#sourcedata

At EMBO Press, we ask authors to provide source data for the main manuscript figures. Our source data coordinator will contact you to discuss which figure panels we would need source data for and will also provide you with helpful tips on how to upload and organize the files.

In the interest of ensuring the conceptual advance provided by the work, we recommend submitting a revision within 3 months (13th Mar 2024). Please discuss the revision progress ahead of this time with the editor if you require more time to complete the revisions. Use the link below to submit your revision:

Link Not Available

Referee #2:

The authors have significantly improved the manuscript, both the data presented and the revised text provide now a much more coherent and convincing analysis of KIF18A sensitivity in a range of cell lines. I am now happy to support publication.

Referee #3:

Within the revised manuscript, the authors have made a significant effort to address the concerns raised by multiple reviewers about generalized conclusions. Given this, and the large amount of useful data included in the paper, I believe the paper is appropriate for publication. I do encourage the authors to reconsider two of the original concerns I raised, which are both related to using more precise language when describing results.

1. The authors did not directly address, in their response to reviewers, my concern that their data do not support the conclusion that weakened APC/C activity "drives" vulnerability to KIF18A. This concern still stands. In fact, the data in Figure 5E demonstrate that inhibiting APC4 in RPE1 cells does not induce sensitivity to KIF18A inhibition. However, vulnerability is increased in 4N RPE1 cells after APC4 KD. Thus, reduced APC activity can contribute to KIF18Ai sensitivity, but doesn't "drive" it on its own, at least under the conditions tested. Furthermore, the new analyses of cyclin B degradation in OVCAR-8 and HCC1806 cells suggest that the metaphase delay in those cell types is due to a defect in SAC silencing rather than low basal APC/C activity. This is an important distinction because it indicates a variety of defects could contribute to sensitivity. This idea is reflected in the model in Figure 7 but is not consistent with some language used in the text and title of the manuscript.

2. In response to my previous question about quantifying chromosome oscillations, which describe a distinct pattern of movement, the authors explain that they did not use an imaging time resolution that allowed tracking of individual chromosomes and instead looked at the pattern of chromosome positions every 5 min and inferred that defects were caused by increased oscillation amplitudes. However, this distinction is not reflected in the results text. To avoid confusion, I suggest improving the clarity of the explanation and avoiding statements like "KIF18Ai treatment also caused chromosome hyperoscillation" unless this was actually measured. The term "hyperoscillation" could also describe either increased oscillation amplitudes, which I think the authors are intending here, or decreased oscillation periods (i.e. changing directions more often). Therefore, the term is neither appropriate nor precise enough to describe the images presented in Figure 3E, which lack quantification. Also, the one sentence added to the methods to describe the experiment does not contain enough detail to determine what was actually done: "The degree of chromosome hyperoscillation during mitosis was subjectively evaluated using and standardized using a set of reference images."

All editorial and formatting issues were resolved by the authors.

Dr. Andrew J Holland
Johns Hopkins School of Medicine
Dept of Molecular Biology & Genetics
725 N Wolfe St, 704 PCTB
Baltimore, MD 21205

2nd Jan 2024

Re: EMBOJ-2023-114411R1
Weakened APC/C activity at mitotic exit drives cancer vulnerability to KIF18A inhibition

Dear Andrew,

Thank you for submitting your final revised manuscript for our consideration. I am pleased to inform you that we have now accepted it for publication in The EMBO Journal.

Yours sincerely,

Hartmut
